# Contrastive Learning for Unsupervised Domain Adaptation of Time Series

**Yilmazcan Ozyurt**
ETH Zürich
yozyurt@ethz.ch

**Stefan Feuerriegel**
LMU Munich
feuerriegel@lmu.de

**Ce Zhang**
ETH Zürich
ce.zhang@inf.ethz.ch

## Abstract

Unsupervised domain adaptation (UDA) aims at learning a machine learning model using a labeled source domain that performs well on a similar yet different, unlabeled target domain. UDA is important in many applications such as medicine, where it is used to adapt risk scores across different patient cohorts. In this paper, we develop a novel framework for UDA of time series data, called CLUDA. Specifically, we propose a contrastive learning framework to learn contextual representations in multivariate time series, so that these preserve label information for the prediction task. In our framework, we further capture the variation in the contextual representations between source and target domain via a custom nearest-neighbor contrastive learning. To the best of our knowledge, ours is the first framework to learn domain-invariant, contextual representation for UDA of time series data. We evaluate our framework using a wide range of time series datasets to demonstrate its effectiveness and show that it achieves state-of-the-art performance for time series UDA.

## 1 Introduction

Many real-world applications of machine learning are characterized by differences between the domains at training and deployment (Hendrycks & Dietterich, 2019; Koh et al., 2021). Therefore, effective methods are needed that learn domain-invariant representations across domains. For example, it is well known that medical settings suffer from substantial domain shifts due to differences in patient cohorts, medical routines, reporting practices, etc. (Futoma et al., 2020; Zech et al., 2018). Hence, a machine learning model trained for one patient cohort may not generalize to other patient cohorts. This highlights the need for effective domain adaptation of time series.

***Unsupervised domain adaptation (UDA)*** aims to learn a machine learning model using a labeled source domain that performs well on a similar yet different, unlabeled target domain (Ganin et al., 2016; Long et al., 2018). So far, many methods for UDA have been proposed for *computer vision* (Chen et al., 2020a; Ganin et al., 2016; Huang et al., 2021; Kang et al., 2019; Long et al., 2018; Pei et al., 2018; Shu et al., 2018; Singh, 2021; Sun & Saenko, 2016; Tang et al., 2021; Tzeng et al., 2014; 2017; Xu et al., 2020; Zhu et al., 2020). These works can – in principle – be applied to time series (with some adjustment of their feature extractor); however, they are not explicitly designed to fully leverage time series properties.

In contrast, comparatively few works have focused on UDA of *time series*. Here, previous works utilize a tailored feature extractor to capture temporal dynamics of multivariate time series, typically through recurrent neural networks (RNNs) (Purushotham et al., 2017), long short-term memory (LSTM) networks (Cai et al., 2021), and convolutional neural networks (Liu & Xue, 2021; Wilson et al., 2020; 2021). Some of these works minimize the domain discrepancy of learned features via adversarial-based methods (Purushotham et al., 2017; Wilson et al., 2020; 2021; Jin et al., 2022) or restrictions through metric-based methods (Cai et al., 2021; Liu & Xue, 2021).

Another research stream has developed time series methods for *transfer learning* from the source domain to the target domain (Eldele et al., 2021; Franceschi et al., 2019; Kiyasseh et al., 2021; Tonekaboni et al., 2021; Yang & Hong, 2022; Yèche et al., 2021; Yue et al., 2022). These methods pre-train a neural network model via contrastive learning to capture the contextual representation of time series from unlabeled source domain. However, these methods operate on a labeled target

domain, which is different from UDA. To the best of our knowledge, there is no method for UDA of time series that captures and aligns the contextual representation across source and target domains.

In this paper, we propose a novel framework for unsupervised domain adaptation of time series data based on contrastive learning, called CLUDA. Different from existing works, our CLUDA framework aims at capturing the contextual representation in multivariate time series as a form of high-level features. To accomplish this, we incorporate the following components: (1) We minimize the domain discrepancy between source and target domains through adversarial training. (2) We capture the contextual representation by generating positive pairs via a set of semantic-preserving augmentations and then learning their embeddings. For this, we make use of contrastive learning (CL). (3) We further align the contextual representation across source and target domains via a custom nearest-neighborhood contrastive learning.

We evaluate our method using a wide range of time series datasets. (1) We conduct extensive experiments using established **benchmark datasets** WISDM (Kwapisz et al., 2011), HAR (Anguita et al., 2013), and HHAR (Stisen et al., 2015). Thereby, we show our CLUDA leads to increasing accuracy on target domains by an important margin. (2) We further conduct experiments on two large-scale, *real-world* **medical datasets**, namely MIMIC-IV (Johnson et al., 2020) and AmsterdamUMCdb (Thoral et al., 2021). We demonstrate the effectiveness of our framework for our medical setting and confirm its superior performance over state-of-the-art baselines. In fact, medical settings are known to suffer from substantial domain shifts across health institutions (Futoma et al., 2020; Nestor et al., 2019; Zech et al., 2018). This highlights the relevance and practical need for adapting machine learning across domains from training and deployment.

**Contributions:**[1]

1. We propose a novel, contrastive learning framework (CLUDA) for unsupervised domain adaptation of time series. To the best of our knowledge, ours is the first UDA framework that learns a contextual representation of time series to preserve information on labels.

2. We capture domain-invariant, contextual representations in CLUDA through a custom approach combining nearest-neighborhood contrastive learning and adversarial learning to align them across domains.

3. We demonstrate that our CLUDA achieves state-of-the-art performance. Furthermore, we show the practical value of our framework using large-scale, real-world medical data from intensive care units.

## 2 RELATED WORK

**Contrastive learning:** Contrastive learning aims to learn representations with self-supervision, so that similar samples are embedded close to each other (positive pair) while pushing dissimilar samples away (negative pairs). Such representations have been shown to capture the semantic information of the samples by maximizing the lower bound of the mutual information between two augmented views (Bachman et al., 2019; Tian et al., 2020a;b). Several methods for contrastive learning have been developed so far (Oord et al., 2018; Chen et al., 2020b; Dwibedi et al., 2021; He et al., 2020), and several of which are tailored to unsupervised representation learning of time series (Franceschi et al., 2019; Yèche et al., 2021; Yue et al., 2022; Tonekaboni et al., 2021; Kiyasseh et al., 2021; Eldele et al., 2021; Yang & Hong, 2022; Zhang et al., 2022). A detailed review is in Appendix A.

**Unsupervised domain adaptation:** Unsupervised domain adaptation leverages labeled source domain to predict the labels of a different, unlabeled target domain (Ganin et al., 2016). To achieve this, UDA methods typically aim to minimize the domain discrepancy and thereby decrease the lower bound of the target error (Ben-David et al., 2010). To minimize the domain discrepancy, existing UDA methods can be loosely grouped into three categories: (1) *Adversarial-based methods* reduce the domain discrepancy via domain discriminator networks, which enforce the feature extractor to learn domain-invariant feature representations. Examples are DANN (Ganin et al., 2016), CDAN (Long et al., 2018), ADDA (Tzeng et al., 2017), MADA (Pei et al., 2018), DIRT-T (Shu et al., 2018), and DM-ADA (Xu et al., 2020). (2) *Contrastive methods* reduce the domain discrepancy through a minimization of a contrastive loss which aims to bring source and target embeddings of the same class

---

[1]Codes are available at https://github.com/oezyurty/CLUDA .

together. Here, the labels (i.e., class information) of the target samples are unknown, and, hence, these methods rely on pseudo-labels of the target samples generated from a clustering algorithm, which are noisy estimates of the actual labels of the target samples. Examples are CAN (Kang et al., 2019), CLDA (Singh, 2021), GRCL (Tang et al., 2021), and HCL (Huang et al., 2021). (3) *Metric-based methods* reduce the domain discrepancy by enforcing restrictions through a certain distance metric (e.g., via regularization). Examples are DDC (Tzeng et al., 2014), Deep CORAL (Sun & Saenko, 2016), DSAN (Zhu et al., 2020), HoMM (Chen et al., 2020a), and MMDA (Rahman et al., 2020). However, previous works on UDA typically come from computer vision. There also exist works on UDA for videos (e. g., Sahoo et al., 2021); see Appendix A for details. Even though these methods can be applied to time series through tailored feature extractors, they do not fully leverage the time series properties. In contrast, comparatively few works have been proposed for UDA of time series.

**Unsupervised domain adaptation for time series:** A few methods have been tailored to unsupervised domain adaptation for time series data. Variational recurrent adversarial deep domain adaptation (VRADA) (Purushotham et al., 2017) was the first UDA method for multivariate time series that uses adversarial learning for reducing domain discrepancy. In VRADA, the feature extractor is a variational recurrent neural network (Chung et al., 2015), and VRADA then trains the classifier and the domain discriminator (adversarially) for the last latent variable of its variational recurrent neural network. Convolutional deep domain adaptation for time series (CoDATS) (Wilson et al., 2020) builds upon the same adversarial training as VRADA, but uses convolutional neural network for the feature extractor instead. Time series sparse associative structure alignment (TS-SASA) (Cai et al., 2021) is a metric-based method. Here, intra-variables and inter-variables attention mechanisms are aligned between the domains via the minimization of maximum mean discrepancy (MMD). Adversarial spectral kernel matching (AdvSKM) (Liu & Xue, 2021) is another metric-based method that aligns the two domains via MMD. Specifically, it introduces a spectral kernel mapping, from which the output is used to minimize MMD between the domains. Across all of the aforementioned methods, the aim is to align the features across source and target domains.

**Research gap:** For UDA of time series, existing works merely align the *features* across source and target domains. Even though the source and target distributions overlap, this results in mixing the source and target samples of different classes. In contrast to that, we propose to align the *contextual representation*, which preserves the label information. This facilitates a better alignment across domains for each class, leading to a better generalization over unlabeled target domain. To achieve this, we develop a novel framework called CLUDA based on contrastive learning.

## 3   PROBLEM DEFINITION

We consider a classification task for which we aim to perform *UDA of time* series. Specifically, we have two distributions over the time series from the source domain $\mathcal{D}_S$ and the target domain $\mathcal{D}_t$. In our setup, we have **labeled** *i.i.d.* samples from the source domain given by $\mathcal{S} = \{(x_i^s, y_i^s)\}_{i=1}^{N_s} \sim \mathcal{D}_S$, where $x_i^s$ is a sample of the source domain, $y_i^s$ is the label for the given sample, and $N_s$ is the overall number of *i.i.d.* samples from the source domain. In contrast, we have **unlabeled** *i.i.d.* samples from the target domain given by $\mathcal{T} = \{x_i^t\}_{i=1}^{N_t} \sim \mathcal{D}_T$, where $x_i^t$ is a sample of the target domain and $N_t$ is the overall number of *i.i.d.* samples from the target domain.

In this paper, we allow for multivariate time series. Hence, each $x_i$ (either from the source or target domain) is a sample of multivariate time series denoted by $x_i = \{x_{it}\}_{t=1}^T \in \mathbb{R}^{M \times T}$, where $T$ is the number of time steps and $x_{it} \in \mathbb{R}^M$ is $M$ observations for the corresponding time step.

Our aim is to build a classifier that generalizes well over target samples $\mathcal{T}$ by leveraging the labeled source samples $\mathcal{S}$. Importantly, labels for the target domain are **not** available during training. Instead, we later use the labeled target samples $\mathcal{T}_{\text{test}} = \{(x_i^t, y_i^t)\}_{i=1}^{N_{\text{test}}} \sim \mathcal{D}_T$ only for the evaluation.

The above setting is directly relevant for practice (Futoma et al., 2020; Hendrycks & Dietterich, 2019; Koh et al., 2021; Zech et al., 2018). For example, medical time series from different health institutions differ in terms of patient cohorts, medical routines, reporting practice, etc., and, therefore, are subject to substantial domain shifts. As such, data from training and data from deployment should be considered as different domains. Hence, in order to apply machine learning for risk scoring or other medical use cases, it is often helpful to adapt the machine learning model trained on one domain $\mathcal{S}$ for another domain $\mathcal{T}$ before deployment.

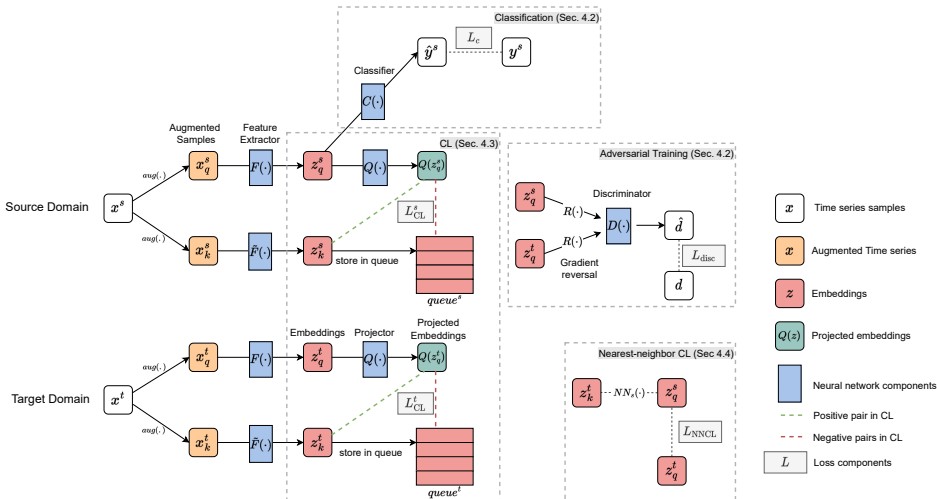

Figure 1: The complete CLUDA framework (best viewed in color). Some network components are shown twice (for source and target) to enhance readability. Source and target samples are augmented twice (colored in yellow). These augmented samples are processed by the feature extractor to yield the embeddings (colored in red). The embeddings are processed by four different components: classification network (Sec. 4.2), adversarial training (Sec. 4.2), CL (Sec. 4.3), and nearest-neighbor CL (Sec. 4.4). Dashed lines represent input pairs to each loss function.

## 4 PROPOSED CLUDA FRAMEWORK

In this section, we describe the components of our framework to learn domain-invariant, contextual representation of time series. We start with an overview of our CLUDA framework, and then describe how we (1) perform domain adversarial training, (2) capture the contextual representation, and (3) align contextual representation across domains.

### 4.1 ARCHITECTURE

The neural architecture of our CLUDA for unsupervised domain adaptation of time series is shown in Fig. 1. In brief, our architecture is the following. (1) The **feature extractor** network $F(\cdot)$ takes the (augmented) time series $x^s$ and $x^t$ from both domains and creates corresponding embeddings $z^s$ and $z^t$, respectively. The **classifier network** $C(\cdot)$ is trained to predict the label $y^s$ of time series from the source domain using the embeddings $z^s$. The **discriminator network** $D(\cdot)$ is trained to distinguish source embeddings $z^s$ from target embeddings $z^t$. For such training, we introduce domain labels $d = 0$ for source instances and $d = 1$ for target instances. The details of how classifier and discriminator have been trained is explained in Sec. 4.2. Note that we later explicitly compare our CLUDA against this base architecture based on "standard" domain adversarial learning. We refer to it as "w/o CL and w/o NNCL". (2) Our CLUDA further captures the contextual representation of the time series in the embeddings $z^s$ and $z^t$. For this, our framework leverages the **momentum updated feature extractor** network $\tilde{F}(\cdot)$ and the **projector network** $Q(\cdot)$ via contrastive learning for each domain. The details are described in Sec. 4.3. (3) Finally, CLUDA aligns the contextual representation across domains in the embedding space $z^s$ and $z^t$ via nearest-neighbor CL. This is explained in Sec. 4.4. The overall training objective of CLUDA is given in Sec. 4.5.

### 4.2 ADVERSARIAL TRAINING FOR UNSUPERVISED DOMAIN ADAPTATION

For the adversarial training, we minimize a combination of two losses: (1) Our *prediction loss* $L_c$ trains the feature extractor $F(\cdot)$ and the classifier $C(\cdot)$. We train both jointly in order to correctly predict the labels from the source domain. For this, we define the prediction loss

$$L_c = \frac{1}{N_s} \sum_i^{N_s} L_{\text{pred}}(C(F(x_i^s)), y_i^s), \tag{1}$$

where $L_{\text{pred}}(\cdot, \cdot)$ is the cross-entropy loss.

(2) Our *domain classification loss* $L_{\text{disc}}$ is used to learn domain-invariant feature representations. Here, we use adversarial learning (Ganin et al., 2016). To this end, the domain discriminator $D(\cdot)$ is trained to minimize the domain classification loss, whereas the feature extractor $F(\cdot)$ is trained to maximize the same loss simultaneously. This is achieved by the gradient reversal layer $R(\cdot)$ between $F(\cdot)$ and $D(\cdot)$, defined by

$$R(x) = x, \qquad \frac{\mathrm{d}R}{\mathrm{d}x} = -\mathbf{I}. \tag{2}$$

Hence, we yield the domain classification loss

$$L_{\text{disc}} = \frac{1}{N_s} \sum_i^{N_s} L_{\text{pred}}(D(R(F(x_i^s))), d_i^s) + \frac{1}{N_t} \sum_i^{N_t} L_{\text{pred}}(D(R(F(x_i^t))), d_i^t). \tag{3}$$

### 4.3 CAPTURING CONTEXTUAL REPRESENTATIONS

In our CLUDA, we capture a contextual representation of the time series in the embeddings $z^s$ and $z^t$, and then align the contextual representations of the two domains for unsupervised domain adaptation. With this approach, we improve upon the earlier works in two ways: (1) We encourage our feature extractor $F(\cdot)$ to learn label-preserving information captured by the context. This observation was made earlier for unsupervised representation learning yet outside of our time series settings (Bachman et al., 2019; Chen et al., 2020b;c; Ge et al., 2021; Grill et al., 2020; Tian et al., 2020a;b). (2) We further hypothesize that discrepancy between the contextual representations of two domains is smaller than discrepancy between their feature space, therefore, the domain alignment task becomes easier.

To capture the contextual representations of time series for each domain, we leverage contrastive learning. CL is widely used in unsupervised representation learning for the downstream tasks in machine learning (Chen et al., 2020b;c; He et al., 2020; Mohsenvand et al., 2020; Shen et al., 2022; Yèche et al., 2021; Zhang et al., 2022). In plain words, CL approach aims to learn similar representations for two augmented views (positive pair) of the same sample in contrast to the views from other samples (negative pairs). This leads to maximizing the mutual information between two views and, therefore, capturing contextual representation (Bachman et al., 2019; Tian et al., 2020a;b).

In our framework (see Fig. 1), we leverage contrastive learning in form of momentum contrast (MoCo) (He et al., 2020) in order to capture the contextual representations from each domain. Accordingly, we apply semantic-preserving augmentations (Cheng et al., 2020; Kiyasseh et al., 2021; Yèche et al., 2021) to each sample of multivariate time series twice. Specifically, in our framework, we sequentially apply the following functions with random instantiations: history crop, history cutout, channel dropout, and Gaussian noise (see Appendix C for details). After augmentation, we have two views of each sample, called query $x_q$ and key $x_k$. These two views are then processed by the feature extractor to get their embeddings as $z_q = F(x_q)$ and $z_k = \tilde{F}(x_k)$. Here, $\tilde{F}(\cdot)$ is a *momentum-updated* feature extractor for MoCo.

To train the momentum-updated feature extractor, the gradients are not backpropagated through $\tilde{F}(\cdot)$. Instead, the weights $\theta_{\tilde{F}}$ are updated by the momentum via

$$\theta_{\tilde{F}} \leftarrow m\,\theta_{\tilde{F}} + (1 - m)\,\theta_F, \tag{4}$$

where $m \in [0, 1)$ is the momentum coefficient. The objective of MoCo-based contrastive learning is to project $z_q$ via a projector network $Q(\cdot)$ and bring the projection $Q(z_q)$ closer to its positive sample $z_k$ (as opposed to negative samples stored in queue $\{z_{kj}\}_{j=1}^J$), which is a collection of $z_k$'s from the earlier batches. This generates a large set of negative pairs (queue size $J \gg$ batch size $N$), which, therefore, facilitates better contextual representations (Bachman et al., 2019; Tian et al., 2020a;b). After each training step, the batch of $z_k$'s are stored in queue of size $J$. As a result, for each domain, we have the following *contrastive loss*

$$L_{\text{CL}} = -\frac{1}{N} \sum_{i=1}^N \log \frac{\exp(Q(z_{qi}) \cdot z_{ki}/\tau)}{\exp(Q(z_{qi}) \cdot z_{ki}/\tau) + \sum_{j=1}^J \exp(Q(z_{qi}) \cdot z_{kj}/\tau)}, \tag{5}$$

where $\tau > 0$ is the temperature scaling parameter, and where all embeddings are normalized. Since we have two domains (i.e., source and target), we also have two contrastive loss components given by $L_{\text{CL}}^s$ and $L_{\text{CL}}^t$ and two queues given by $queue^s$ and $queue^t$, respectively.

## 4.4 Aligning the Contextual Representation Across Domains

Our CLUDA framework further aligns the contextual representation across the source and target domains. To do so, we build upon ideas for nearest-neighbor contrastive learning (Dwibedi et al., 2021) from unsupervised representation learning, yet outside of our time series setting. To the best of our knowledge, ours is the first nearest-neighbor contrastive learning approach for unsupervised domain adaptation of time series.

In our CLUDA framework, nearest-neighbor contrastive learning (NNCL) should facilitate the classifier $C(\cdot)$ to make accurate predictions for the target domain. We achieve this by creating positive pairs between domains, whereby we explicitly align the representations across domains. For this, we pair $z_{qi}^t$ with the nearest-neighbor of $z_{ki}^t$ from the source domain, denoted as $NN_s(z_{ki}^t)$. We thus introduce our *nearest-neighbor contrastive learning loss*

$$L_{\text{NNCL}} = -\frac{1}{N_t} \sum_{i=1}^{N_t} \log \frac{\exp(z_{qi}^t \cdot NN_s(z_{ki}^t)/\tau)}{\sum_{j=1}^{N_s} \exp(z_{qi}^t \cdot z_{qj}^s/\tau)}, \tag{6}$$

where $NN_s(\cdot)$ retrieves the nearest-neighbor of an embedding from the source queries $\{z_{qi}^s\}_{i=1}^{N_s}$.

## 4.5 Training

**Overall loss:** Overall, our CLUDA framework minimizes

$$L = L_c + \lambda_{\text{disc}} \cdot L_{\text{disc}} + \lambda_{\text{CL}} \cdot (L_{\text{CL}}^s + L_{\text{CL}}^t) + \lambda_{\text{NNCL}} \cdot L_{\text{NNCL}}, \tag{7}$$

where hyperparameters $\lambda_{\text{disc}}$, $\lambda_{\text{CL}}$, and $\lambda_{\text{NNCL}}$ control the contribution of each component.

**Implementation:** Appendix C provides all details of our architecture search for each component.

## 5 Experimental Setup

Our evaluation is two-fold: (1) We conduct extensive experiments using established **benchmark datasets**, namely WISDM (Kwapisz et al., 2011), HAR (Anguita et al., 2013), and HHAR (Stisen et al., 2015). Here, sensor measurements of each participant are treated as separate domains and we randomly sample 10 source-target domain pairs for evaluation. This has been extensively used in the earlier works of UDA on time series (Wilson et al., 2020; 2021; Cai et al., 2021; Liu & Xue, 2021). Thereby, we show how our CLUDA leads to increasing accuracy on target domains by an important margin. (2) We show the applicability of our framework in a *real-world* setting with **medical datasets**: MIMIC-IV (Johnson et al., 2020) and AmsterdamUMCdb (Thoral et al., 2021). These are the largest intensive care units publicly available and both have a different origin (Boston, United States vs. Amsterdam, Netherlands). Therefore, they reflect patients with different characteristics, medical procedures, etc. Here we treat each age group as a separate domain (Purushotham et al., 2017; Cai et al., 2021). Further details of datasets and task specifications are in Appendix B.

**Baselines:** (1) We report the performance of a model without UDA (**w/o UDA**) to show the overall contribution of UDA methods. For this, we only use feature extractor $F(\cdot)$ and the classifier $C(\cdot)$ using the same architecture as in our CLUDA. This model is only trained on the source domain. (2) We implement the following state-of-the-art baselines for UDA of time series data. These are: **VRADA** (Purushotham et al., 2017), **CoDATS** (Wilson et al., 2020), **TS-SASA** (Cai et al., 2021), and **AdvSKM** (Liu & Xue, 2021). In our results later, we omitted TS-SASA as it repeatedly was not better than random. (3) We additionally implement **CAN** (Kang et al., 2019), **CDAN** (Long et al., 2018), **DDC** (Tzeng et al., 2014), **DeepCORAL** (Sun & Saenko, 2016), **DSAN** (Zhu et al., 2020), **HoMM** (Chen et al., 2020a), and **MMDA** (Rahman et al., 2020). These models were originally developed for computer vision, but we tailored their feature extractor to time series (See Appendix C).

## 6 Results

### 6.1 Prediction Performance on Benchmark Datasets

Figure 2a shows the average accuracy of each method for 10 source-target domain pairs on the WISDM, HAR, and HHAR datasets. On WISDM dataset, our CLUDA outperforms the best baseline

accuracy of CoDATS by 12.7 % (0.754 vs. 0.669). On HAR dataset, our CLUDA outperforms the best baseline accuracy of CDAN by 18.9 % (0.944 vs. 0.794). On HHAR dataset, our CLUDA outperforms the best baseline accuracy of CDAN by 21.8 % (0.759 vs. 0.623). Overall, CLUDA consistently improves the best UDA baseline performance by a large margin. In Appendix D, we provide the full list of UDA results for each source-target pair, and additionally provide Macro-F1 scores, which confirm our findings from above.

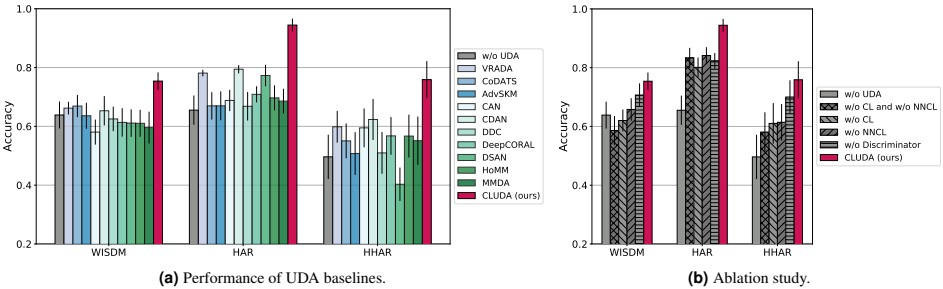

(a) Performance of UDA baselines.    (b) Ablation study.

Figure 2: UDA performance on benchmark datasets.

**Insights:** We further visualize the embeddings in Fig. 3 to study the domain discrepancy and how our CLUDA aligns the representation of time series. (a) The embeddings of w/o UDA show that there is a significant domain shift between source and target. This can be observed by the two clusters of each class (i. e., one for each domain). (b) CDAN as the best baseline reduces the domain shift by aligning the features of source and target for some classes, yet it mixes the different classes of the different domains (e.g., blue class of source and green class of target overlap). (c) By examining the embeddings from our CLUDA, we confirm its effectiveness: (1) Our CLUDA pulls together the source (target) classes for the source (target) domain (due to the CL). (2) Our CLUDA further pulls both source and target domains together for each class (due to the alignment).

We have the following observation when we consider the embedding visualizations of all baselines (see Appendix E). Overall, all the baselines show certain improvements over w/o UDA in aligning the embedding distributions of the source and target domains (i. e., overlapping point clouds of source and target domains). Yet, when the class-specific embedding distributions are considered, source and target samples are fairly separated. Our CLUDA remedies this issue by actively pulling source and target samples of the same class together via its novel components.

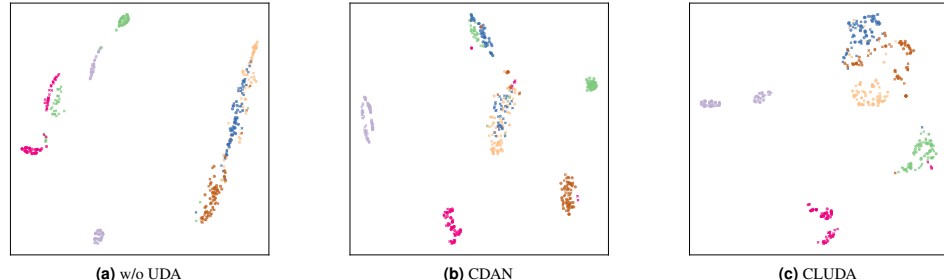

(a) w/o UDA    (b) CDAN    (c) CLUDA

Figure 3: t-SNE visualization for the embeddings from HHAR dataset. Each class is represented by a different color. Shape shows source and target domains (circle vs. cross).

**Ablation study:** We conduct an ablation study (see Fig. 2b) to better understand the importance of the different components in our framework. The variants are: (1) **w/o UDA** baseline for comparison; (2) **w/o CL and w/o NNCL**, which solely relies on the domain adversarial training and refers to base architecture from Sec. 4.2; (3) **w/o CL**, which deactivates CL (from Sec. 4.3) for capturing contextual representations; (4) **w/o NNCL**, which deactivates NNCL (from Sec. 4.4) for aligning contextual representations across domains; and (5) **w/o Discriminator**, which deactivates the discriminator.

Overall, the low prediction performance of the adversarial training from Sec. 4.2 (**w/o CL and w/o NNCL**) demonstrates the importance of *capturing* and *aligning* the contextual representations. Comparing **w/o Discriminator** shows that the largest part of our performance improvements (compared to **w/o UDA**) comes from our novel components introduced in Sec. 4.3 and Sec. 4.4 to capture

and align the contextual representation of time series. The discriminator also helps achieving consistent performance gains, albeit of smaller magnitude. Finally, our CLUDA works the best in all experiments, thereby justifying our chosen architecture. Appendix F shows our detailed ablation study. We further conduct an ablation study to understand the importance of the selected CL method. For this, we implement two new variants of our CLUDA: (1) CLUDA with SimCLR (Chen et al., 2020b) and (2) CLUDA with NCL (Yèche et al., 2021). Both variants performed inferior, showing the importance of choosing a tailored CL method for time series UDA. Details are in Appendix K.

## 6.2 Prediction Performance on Medical Datasets

We further demonstrate that our CLUDA achieves state-of-the-art UDA performance using the medical datasets. For this, Table 1 lists 12 UDA scenarios created from the MIMIC dataset. In 9 out of 12 domain UDA scenarios, CLUDA yields the best mortality prediction in the target domain, consistently outperforming the UDA baselines. When averaging over all scenarios, our CLUDA improves over the performance of the best baseline (AdvSKM) by 2.2 % (AUROC 0.773 vs. 0.756). Appendix H reports the ablation study for this experimental setup with similar findings as above.

Appendix G repeats the experiments for another medical dataset: AmsterdamUMCdb (Thoral et al., 2021). Again, our CLUDA achieves state-of-the-art performance.

Table 1: Prediction performance for medical dataset. Task: mortality prediction between various age groups of MIMIC-IV. Shown: mean AUROC over 10 random initializations.

| Sour ↦ Tar | w/o UDA | VRADA | CoDATS | AdvSKM | CAN | CDAN | DDC | DeepCORAL | DSAN | HoMM | MMDA | CLUDA (ours) |
|---|---|---|---|---|---|---|---|---|---|---|---|---|
| 1 ↦ 2 | 0.744 | 0.786 | 0.744 | 0.757 | 0.757 | 0.726 | 0.745 | 0.728 | 0.756 | 0.742 | 0.726 | **0.798** |
| 1 ↦ 3 | 0.685 | 0.729 | 0.685 | 0.702 | 0.687 | 0.654 | 0.694 | 0.688 | 0.701 | 0.654 | 0.684 | **0.747** |
| 1 ↦ 4 | 0.617 | 0.631 | 0.616 | 0.619 | 0.607 | 0.580 | 0.613 | 0.595 | 0.620 | 0.587 | 0.622 | **0.649** |
| 2 ↦ 1 | 0.818 | 0.828 | 0.822 | 0.835 | 0.804 | 0.842 | 0.821 | 0.824 | 0.821 | 0.820 | 0.825 | **0.856** |
| 2 ↦ 3 | 0.790 | 0.746 | **0.797** | 0.792 | 0.789 | 0.788 | 0.791 | 0.789 | 0.795 | **0.797** | 0.793 | 0.796 |
| 2 ↦ 4 | 0.696 | 0.649 | **0.699** | 0.696 | 0.666 | 0.620 | 0.693 | **0.699** | 0.690 | 0.694 | 0.694 | 0.697 |
| 3 ↦ 1 | 0.787 | 0.808 | 0.788 | 0.798 | 0.800 | 0.754 | 0.796 | 0.797 | 0.790 | 0.796 | 0.803 | **0.822** |
| 3 ↦ 2 | 0.833 | 0.805 | 0.832 | 0.835 | 0.837 | 0.777 | 0.831 | 0.827 | 0.833 | 0.834 | 0.830 | **0.843** |
| 3 ↦ 4 | **0.751** | 0.684 | 0.748 | 0.745 | 0.727 | 0.689 | 0.748 | 0.746 | 0.750 | 0.733 | 0.745 | 0.745 |
| 4 ↦ 1 | 0.783 | 0.790 | 0.783 | 0.788 | 0.792 | 0.747 | 0.778 | 0.768 | 0.766 | 0.774 | 0.754 | **0.807** |
| 4 ↦ 2 | 0.761 | 0.760 | 0.762 | 0.765 | 0.765 | 0.748 | 0.761 | 0.740 | 0.757 | 0.756 | 0.744 | **0.769** |
| 4 ↦ 3 | 0.736 | 0.723 | 0.737 | 0.734 | 0.731 | 0.730 | 0.739 | 0.734 | 0.735 | 0.742 | 0.738 | **0.748** |
| Avg | 0.750 | 0.745 | 0.751 | 0.756 | 0.747 | 0.721 | 0.751 | 0.745 | 0.751 | 0.744 | 0.747 | **0.773** |

Higher is better. Best value in bold. Second best results are underlined if std. dev. overlap.

## 6.3 Case Study: Application to Medical Practice

We now provide a case study showing the application of our framework to medical practice. Here, we evaluate the domain adaptation between two health institutions. We intentionally chose this setting as medical applications are known to suffer from a substantial domain shifts (e. g., due to different patient cohorts, medical routines, reporting practices, etc., across health institutions) (Futoma et al., 2020; Nestor et al., 2019; Zech et al., 2018). We treat MIMIC and AmsterdamUMCdb (AUMC) as separate domains and then predict health outcomes analogous to earlier works (Cai et al., 2021; Che et al., 2018; Ge et al., 2018; Ozyurt et al., 2021; Purushotham et al., 2017): decompensation, mortality, and length of stay (see Table 2). All details regarding the medical datasets and task definitions are given in Appendix B.

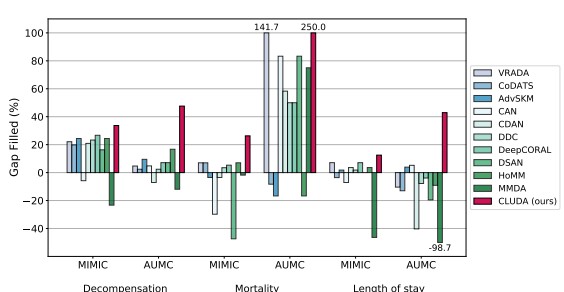

Figure 4: Case study. Report is how much of the performance gap is filled by each method. Here: performance gap [%] is the difference between no domain adaptation and the source ↦ source setting as a loose upper bound on performance.

Table 2: 3 UDA tasks between MIMIC and AUMC. Shown: Average performance over 10 random initializations.

| Task | Decompensation (AUROC) | | | | Mortality (AUROC) | | | | Length-of-stay (KAPPA) | | | |
|---|---|---|---|---|---|---|---|---|---|---|---|---|
| Source | MIMIC | | AUMC | | MIMIC | | AUMC | | MIMIC | | AUMC | |
| Target | MIMIC | AUMC | AUMC | MIMIC | MIMIC | AUMC | AUMC | MIMIC | MIMIC | AUMC | AUMC | MIMIC |
| w/o UDA | 0.831 | 0.771 | 0.813 | 0.745 | 0.831 | 0.709 | 0.721 | 0.774 | 0.178 | 0.169 | 0.246 | 0.122 |
| VRADA | 0.817 | 0.773 | 0.798 | 0.764 | 0.827 | 0.726 | 0.729 | 0.778 | 0.168 | 0.161 | 0.241 | 0.126 |
| CoDATS | 0.825 | 0.772 | 0.818 | 0.762 | 0.832 | 0.708 | 0.724 | 0.778 | 0.174 | 0.159 | 0.243 | 0.120 |
| AdvSKM | 0.824 | 0.775 | 0.817 | 0.766 | 0.830 | 0.707 | 0.724 | 0.772 | 0.179 | 0.172 | 0.244 | 0.123 |
| CAN | 0.825 | 0.773 | 0.807 | 0.740 | 0.830 | 0.719 | 0.715 | 0.757 | 0.142 | 0.173 | 0.233 | 0.118 |
| CDAN | 0.824 | 0.768 | 0.817 | 0.763 | 0.776 | 0.716 | 0.712 | 0.772 | 0.176 | 0.138 | 0.244 | 0.124 |
| DDC | 0.825 | 0.772 | 0.819 | 0.765 | 0.831 | 0.715 | 0.721 | 0.776 | 0.175 | 0.163 | 0.244 | 0.123 |
| DeepCORAL | 0.832 | 0.774 | 0.819 | 0.768 | 0.832 | 0.715 | 0.727 | 0.777 | 0.175 | 0.166 | 0.244 | 0.126 |
| DSAN | 0.831 | 0.774 | 0.808 | 0.759 | 0.832 | 0.719 | 0.721 | 0.747 | 0.175 | 0.154 | 0.246 | 0.122 |
| HoMM | 0.829 | 0.778 | 0.816 | 0.766 | 0.833 | 0.707 | 0.720 | 0.778 | 0.174 | 0.162 | 0.243 | 0.124 |
| MMDA | 0.821 | 0.766 | 0.814 | 0.725 | 0.831 | 0.718 | 0.724 | 0.773 | 0.158 | 0.093 | 0.246 | 0.096 |
| CLUDA (ours) | 0.832 | **0.791** | 0.825 | **0.774** | 0.836 | **0.739** | 0.750 | **0.789** | 0.216 | **0.202** | 0.276 | **0.129** |

Higher is better. Best value in bold. Black font: main results for UDA. Gray font: source ↦ source.

Fig. 4 shows the performance across all three UDA tasks and for both ways (i.e., MIMIC ↦ AUMC and AUMC ↦ MIMIC). For better comparability in practice, we focus on the *"performance gap"*: we interpret the performance from source ↦ source setting as a loose upper bound.[2] We then report how much of the performance gap between no domain adaptation (w/o UDA) vs. the loose upper bound is filled by each method. Importantly, our CLUDA consistently outperforms the state-of-the-art baselines. For instance, in decompensation prediction for AUMC, our CLUDA (AUROC 0.791) fills 47.6 % of the performance gap between no domain adaptation (AUROC 0.771) and loose upper bound from the source ↦ source setting (AUROC 0.813). In contrast, the best baseline model of this task (HoMM) can only fill 16.7 % (AUROC 0.778). Altogether, this demonstrates the effectiveness of our proposed framework. Appendix I reports the detailed results for different performance metrics. Appendix J provides an ablation study. Both support our above findings.

# 7 DISCUSSION

Our CLUDA framework shows a superior performance for UDA of time series when compared to existing works. Earlier works introduced several techniques for aligning source and target domains, mainly via adversarial training or metric-based losses. Even though they facilitate matching source and target distributions (i. e., overlapping point clouds of two domains), they do not explicitly facilitate matching class-specific distributions across domains. To address this, our CLUDA builds upon two strategies, namely *capturing* and *aligning* contextual representations. (1) CLUDA learns class-specific representations for both domains from the feature extractor. This is achieved by CL, which captures label-preserving information from the context, and therefore, enables adversarial training to align the representations of each class across domains. Yet, the decision boundary of the classifier can still miss some of the target domain samples, since the classifier is prone to overfit to the source domain in high dimensional representation space. To remedy this, (2) CLUDA further aligns the individual samples across domains. This is achieved by our NNCL, which brings each target sample closer to its most similar source domain-counterpart. Therefore, during the evaluation, the classifier generalizes well to target representations which are similar to the source representations from training time.

**Conclusion:** In this paper, we propose a novel framework for UDA of time series based on contrastive learning, called CLUDA. To the best of our knowledge, CLUDA is the first approach that learns domain-invariant, contextual representation in multivariate time series for UDA. Further, CLUDA achieves state-of-the-art performance for UDA of time series. Importantly, our two novel components – i.e., our custom CL and our NNCL – yield clear performance improvements. Finally we expect our framework of direct practical value for medical applications where risk scores should be transferred across populations or institutions.

---

[2]In some cases, this loose upper bound can be exceeded by UDA methods. For instance, when there is not enough number of samples in target domain. Then, leveraging a larger and labeled dataset as source domain can yield a superior performance.

ACKNOWLEDGMENTS

Funding via the Swiss National Science Foundation (SNSF) via Grant 186932 is acknowledged.

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

# A    RELATED WORK

**Contrastive learning:** Several methods for contrastive learning have been developed so far. For example, contrastive predictive coding (CPC) (Oord et al., 2018) predicts the next latent variable in contrast to negative samples from its proposal distribution. SimCLR stands for simple framework for CL of visual representations (Chen et al., 2020b). It maximizes the agreement between the embeddings of the two augmented views of the same sample and treats all the other samples in the same batch as negative samples. Nearest-neighbor CL (NNCL) (Dwibedi et al., 2021) creates positive pairs from other samples in the dataset. For this, it takes the embedding of first augmented view and finds its nearest neighbor from the support set. Moment contrast (MoCo) (He et al., 2020) refers to the embeddings of two augmented views as query and key, and then constructs positive pairs for the sample as follows: Key embeddings are generated by a momentum encoder and stored in a queue (whose size is larger than the batch size), while all key embeddings are further used to construct negative pairs for the other sample. Thereby, MoCo generates more negative pairs than the batch size as compared to SimCLR, which is often more efficient in practice.

**Contrastive learning for time series:** Contrastive learning has been used for time series to learn contextual representation of time series in unsupervised settings. As a result, several methods emerged: Scalable representation learning (SRL) (Franceschi et al., 2019), neighborhood contrastive learning (NCL) (Yèche et al., 2021), TS2Vec (Yue et al., 2022), and temporal neighborhood coding (TNC) (Tonekaboni et al., 2021) treat the neighboring windows of the time series as positive pairs and use other windows to construct negative pairs. For this, SRL, NCL, and TS2Vec minimize the triplet loss, contrastive loss, and hierarchical contrastive loss, respectively, while TNC trains a discriminator network to predict neighborhood information.

There are also more specialized methods. For example, contrastive learning of cardiac signals (CLOCKS) (Kiyasseh et al., 2021) leverages spatial invariance and constructs positive pairs from measurements of the different sensors of the same subject. Temporal and contextual contrasting (TS-TCC) (Eldele et al., 2021) is a variant of CPC and maximizes the agreement between strong and weak augmentations of the same sample in an autoregressive model. Bilinear temporal-spectral fusion (BTSF) (Yang & Hong, 2022) constructs the positive pairs via dropout layer applied to the same sample twice and it minimizes a triplet loss for the temporal and spectral features. Time-Frequency Consistency (TF-C) (Zhang et al., 2022) maximizes the agreement between time and frequency embeddings of the same sample.

**Unsupervised domain adaptation for videos:** Several works have been tailored to unsupervised domain adaptation in video domain. Similar to UDA in other domains, some works leveraged adversarial training for the alignment of source and target domains. Specifically, Temporal Attentive Adversarial Adaptation Network (TA$^3$N) (Chen et al., 2019) assigns different weights to the features of source and target during the domain alignment process, where the weights are determined by the entropy of domain classifier. Adversarial bipartite graph (ABG) (Luo et al., 2020) creates a bi-partite graph from the source and target videos for a given batch and leverages a graph neural network to fool the domain classifier. Temporal Co-attention Network (TCoN) (Pan et al., 2020) generates target-aligned source features via co-attention matrix, which is adversarially trained against domain classifier. Multi-Modal Domain Adaptation for Fine-Grained Action Recognition (Munro & Damen, 2020) further leverages multi-model self supervision during the adversarial training. Shuffle and Attend (Choi et al., 2020) is another adversarial-based method, which additionally predicts the order of the video clips to alleviate the background shift across domains. Different from the previous works, Contrast and Mix (CoMix) (Sahoo et al., 2021) does not rely on the adversarial training. Instead, CoMix generates synthetic videos by mixing the background of source (target) domain and the motion of target (source) domain via convex combination and applies contrastive learning, where the videos sharing the same motion treated are treated as positive pairs. This work aligns source (or target) domain with a synthetic domain, where as our CLUDA framework *directly* aligns two domains, without requiring an intermediate synthetic domain generation. Further, CoMix is not straightforward to be applied in time series, since multivariate time series contains correlated features (i.e. univariate time series) which are not separable as background and motion in videos.

# B    DATASET DETAILS

## B.1    BENCHMARK DATASETS

We select three sensor datasets that are most commonly used in the earlier works. In each dataset, participants perform various actions while wearing smartphone and/or smartwatches. Based on the sensor measurements, the task is to predict which action the participant is performing. Table 3 provides summary statistics for all datasets. Below, we provide additional information about each dataset.

**WISDM**. The dataset contains 3-axis accelerometer measurements from 30 participants. The measurements are collected at 20 Hz, and we use non-overlapping segments of 128 time steps to predict the type of the activity of a participant. There are 6 types of activities: walking, jogging, sitting, standing, walking upstairs and walking downstairs. This dataset is particularly challenging due to class imbalance across participants, i. e., there are some participants who did not perform all the activities.

**HAR**. The dataset contains the measurements of 3-axis accelerometer, 3-axis gyroscope, and 3-axis body acceleration from 30 participants. The measurements are collected at 50 Hz, and we use non-overlapping segments of 128 time steps to predict the type of the activity of a participant. There are 6 types of activities: walking, walking upstairs, walking downstairs, sitting, standing, and lying down.

**HHAR**. The dataset contains 3-axis accelerometer measurements from 30 participants. The measurements are collected at 50 Hz, and we use non-overlapping segments of 128 time steps to predict the type of the activity of a participant. There are 6 types of activities: biking, sitting, standing, walking, walking upstairs, and walking downstairs.

Table 3: Summary of the sensor datasets.

|  | #Subjects | #Channels | Length | # Classes | # Training samples | # Val. samples | # Test samples |
|---|---|---|---|---|---|---|---|
| WISDM | 30 | 3 | 128 | 6 | 3870 | 1043 | 1052 |
| HAR | 30 | 9 | 128 | 6 | 7194 | 1542 | 1563 |
| HHAR | 9 | 3 | 128 | 6 | 10336 | 2214 | 2222 |

## B.2    MEDICAL DATASETS

We use MIMIC-IV (Johnson et al., 2020) and AmsterdamUMCdb (Thoral et al., 2021). Both are de-indentified, publicly-available data from intensive care unit stays, where the goal is to predict mortality. To date, MIMIC-IV is the largest public dataset for intensive care units with 49,351 ICU stays; AmsterdamUMCdb contains 19,840 ICU stays. However, both have a different origin (Boston, United States vs. Amsterdam, Netherlands) and thus reflect patients with different characteristics, medical procedures, etc. For the medical datasets, we follow the literature (Purushotham et al., 2017; Cai et al., 2021) and create 4 domains based on patients' age groups: 20–45, 46–65, 66–85, and 85+ years. We then apply UDA for each cross-domain scenario (i. e., from Group 1 $\mapsto$ Group 4 to Group 4 $\mapsto$ Group 3) to predict mortality.

Table 4 shows the summary statistics of both medical datasets MIMIC and AUMC.

Table 5 provides additional details for both datasets MIMIC and AUMC. Both comprise of 41 separate time series, which are then used to predict the outcomes of interest – i.e., decompensation, mortality, and length of stay – via unsupervised domain adaptation.

Table 4: Summary of datasets.

| Name | From | #Patients | #Measurements | Avg. ICU stay (hours) | Mortality (%) |
|---|---|---|---|---|---|
| MIMIC | US | 49,351 | 41 | 72.21 | 9.95 |
| AUMC | Europe | 19,840 | 41 | 100.13 | 8.62 |

Table 5: Descriptions of medical time series and their summary statistics for MIMIC and AUMC

| Measurement | Unit | MIMIC | | AUMC | |
|---|---|---|---|---|---|
| | | Mean | Std. Dev. | Mean | Std. Dev. |
| Albumin | g/dL | 3.053 | 0.697 | 2.121 | 0.575 |
| Alkaline phosphatase | IU/L | 115.384 | 36.044 | 146.459 | 49.948 |
| Alanine aminotransferase | IU/L | 110.340 | 98.699 | 100.352 | 89.697 |
| Aspartate aminotransferase | IU/L | 142.194 | 160.053 | 115.663 | 131.497 |
| Base excess | mEq/L | 0.691 | 1.744 | 1.291 | 1.740 |
| Bicarbonate | mEq/L | 24.613 | 4.815 | 25.815 | 4.753 |
| Total bilirubin | mg/dL | 2.060 | 4.815 | 0.975 | 2.081 |
| Blood urea nitrogen | mg/dL | 28.063 | 23.073 | 33.112 | 24.829 |
| Calcium | mg/dL | 8.398 | 0.746 | 8.121 | 0.773 |
| Calcium ionized | mmol/L | 1.126 | 0.086 | 1.161 | 0.094 |
| Creatine kinase | IU/L | 978.735 | 1402.472 | 755.806 | 1163.575 |
| Chloride | mEq/L | 103.881 | 6.224 | 107.083 | 6.796 |
| Creatinine | mg/dL | 1.364 | 1.336 | 1.214 | 1.001 |
| Diastolic blood pressure | mmHg | 63.513 | 14.857 | 62.199 | 14.037 |
| Endtidal co2 | mmHg | 35.861 | 7.658 | 35.405 | 8.044 |
| Fraction of inspired oxygen | % | 48.213 | 15.95 | 43.345 | 10.106 |
| Glucose | mg/dL | 134.981 | 50.227 | 134.112 | 36.688 |
| Hematocrit | % | 30.850 | 5.736 | 31.108 | 5.230 |
| Hemoglobin | g/dL | 10.178 | 1.983 | 10.354 | 1.740 |
| Heart rate | bpm | 85.249 | 17.935 | 86.609 | 18.858 |
| Prothrombin time/inter. norm. ratio | _ | 1.381 | 0.616 | 1.312 | 0.476 |
| Potassium | mEq/L | 4.088 | 0.552 | 4.141 | 0.453 |
| Lactate | mmol/L | 1.685 | 1.207 | 1.674 | 1.388 |
| Mean arterial pressure | mmHg | 79.481 | 15.368 | 84.444 | 17.149 |
| Magnesium | mg/dL | 2.107 | 0.347 | 2.160 | 0.446 |
| Sodium | mEq/L | 139.079 | 5.037 | 140.119 | 5.324 |
| Co2 partial pressure | mmHg | 41.259 | 9.551 | 41.590 | 8.355 |
| Ph of blood | _ | 7.401 | 0.070 | 7.411 | 0.069 |
| Phosphate | mg/dL | 3.503 | 1.290 | 3.514 | 1.240 |
| Platelet count | K/uL | 213.982 | 127.242 | 245.261 | 160.271 |
| O2 partial pressure | mmHg | 114.686 | 62.778 | 101.140 | 34.704 |
| Partial thromboplastin time | sec | 36.537 | 18.426 | 44.364 | 18.862 |
| Systolic blood pressure | mmHg | 121.044 | 21.61 | 130.308 | 27.870 |
| Temperature | C | 36.949 | 0.634 | 36.600 | 1.095 |
| White blood cell count | K/uL | 12.152 | 8.583 | 13.314 | 7.804 |
| Basophils | % | 0.264 | 0.362 | 0.227 | 0.417 |
| Eosinophils | % | 1.208 | 1.996 | 1.167 | 1.716 |
| Lymphocytes | % | 12.316 | 10.53 | 11.311 | 9.873 |
| Neutrophils | % | 78.296 | 13.358 | 77.720 | 12.778 |
| Prothrombine time | sec | 15.141 | 6.262 | 3.455 | 43.459 |
| Red blood cell count | m/uL | 3.398 | 0.684 | 3.578 | 0.886 |

Table 6 shows the number of patients and the number of samples for each split and each dataset. As a reminder, since we start making the prediction at four hours after the ICU admission, the same patient yields multiple samples when training/testing the models.

**Pre-processing:** We split the patients of each dataset into 3 parts for training/validation/testing (ratio: 70/15/15). We used a stratified split based on the mortality label. We proceeded analogous to previous works for pre-processing (Cai et al., 2021; Che et al., 2018; Ge et al., 2018; Harutyunyan et al., 2019; Ozyurt et al., 2021; Purushotham et al., 2017; 2018; Yèche et al., 2021). Each measurement was sampled to hourly resolution, and missing measurements were filled with forward-filling imputation. We applied standard scaling to each measurement based on the statistics from training set. The remaining missing measurements were filled with zero, which corresponds to mean imputation after

Table 6: Number of patients and samples for each dataset and each split

|  | MIMIC | | | AUMC | | |
| --- | --- | --- | --- | --- | --- | --- |
|  | Train | Validation | Test | Train | Validation | Test |
| Number of patients | 34,290 | 7,343 | 7,353 | 13,802 | 2958 | 2964 |
| Number of samples | 2,398,546 | 513,636 | 512,454 | 1,332,390 | 304,981 | 287,599 |

scaling. We followed best practices in benchmarking data from intensive care units (Harutyunyan et al., 2019; Purushotham et al., 2018). That is, for each of the tasks, we start making the prediction at four hours after the ICU admission. In all our experiments, we used a maximum history length $T = 48$ hours. Shorter sequences were pre-padded by zero.

**Tasks:** We compare the performance of our framework across 3 different standard tasks from the literature (Cai et al., 2021; Che et al., 2018; Ge et al., 2018; Harutyunyan et al., 2019; Ozyurt et al., 2021; Purushotham et al., 2017; 2018). (1) *Decompensation* prediction refers to predicting whether the patient dies within the next 24 hours. (2) *Mortality* prediction refers to predicting whether the patient dies during his/her ICU stay. (3) *Length of stay* prediction refers to predicting the remaining hours of ICU stay for the given patient. This serves as a proxy of the overall health outcome. The distribution of remaining length of ICU stay contains a heavy tail (see Appendix A), which makes it challenging to model it as a regression task. Therefore, we follow the previous works (Harutyunyan et al., 2019; Purushotham et al., 2018) and divide the range of values into 10 buckets and perform an ordinal multiclass classification.

For each task, we performed unsupervised domain adaptation in both ways: MIMIC (source) $\mapsto$ AUMC (target), and AUMC (source) $\mapsto$ MIMIC (target). Later, we also report the corresponding performance on the test samples from the source dataset (i.e., MIMIC $\mapsto$ MIMIC, and AUMC $\mapsto$ AUMC). This way, we aim to provide insights to what extent the different UDA methods provide a trade-off for the performance in the source vs. target domain. It also be loosely interpreted as a upper bound for the prediction performance.

**Performance metrics:** We report the following performance metrics. The tasks for predicting (1) *decompensation* and (2) *mortality* are binary classification problems. For these tasks, we compare the area under the receiver operating characteristics curve (AUROC) and area under the precision-recall curve (AUPRC). Results for AUPRC are in the Appendix C due to space limitation. The task of predicting (3) *length of stay* is an ordinal multiclass classification problem. For this, we report Cohen's linear weighted kappa, which measures the correlation between the predicted and ground-truth classes.

SUMMARY STATISTICS FOR "LENGTH OF STAY"

Here, we provide additional summary statistics for the distribution of "length of stay". Figure 5 and Figure 6 show the length of stay distribution of all patients in the MIMIC and AUMC datasets, respectively. Further, Figure 7 and Figure 8 show the remaining length of stay distribution for all samples (i.e., all time windows considered for all patients) in MIMIC and AUMC, respectively. Recall that we divide the values of remaining length of stay into 10 buckets; the corresponding fraction of samples belonging to each bucket is reported in Figure 9.

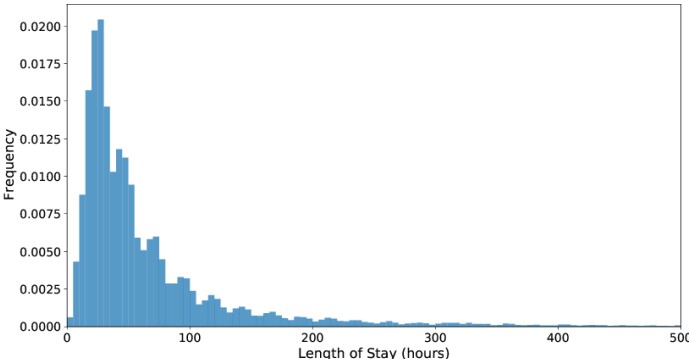

Figure 5: Length of stay distribution of MIMIC patients. For reasons of space, the distribution is cropped at a value of 500.

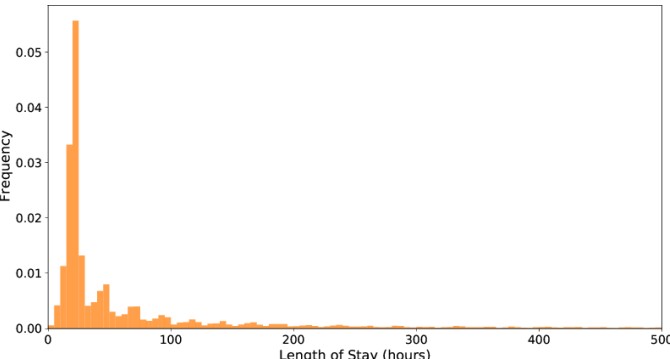

Figure 6: Length of stay distribution of AUMC patients. For reasons of space, the distribution is cropped at a value of 500.

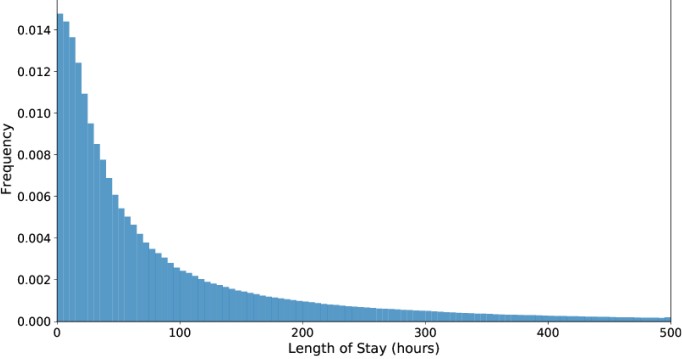

Figure 7: Remaining length of stay distribution of all MIMIC samples. For reasons of space, the distribution is cropped at a value of 500.

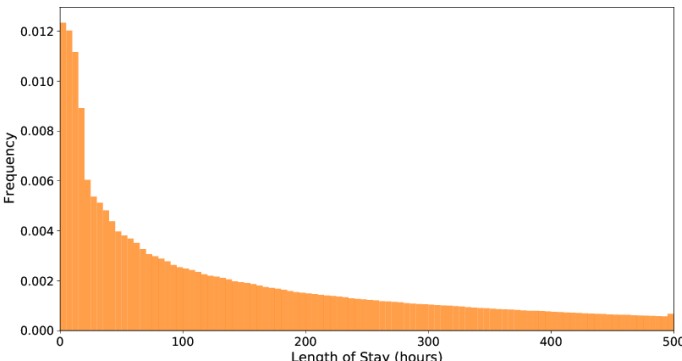

Figure 8: Remaining length of stay distribution of all AUMC samples. TFor reasons of space, the distribution is cropped at a value of 500.

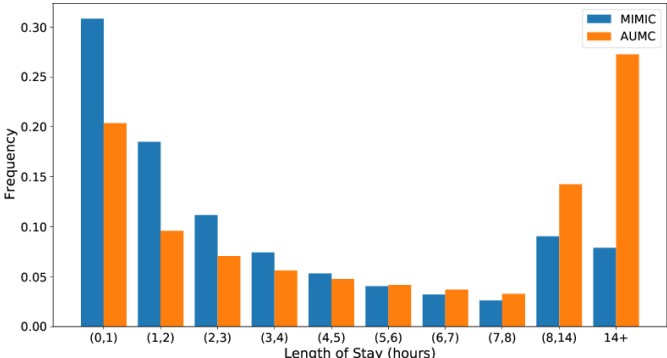

Figure 9: Histogram showing the distribution of remaining length of stay (MIMIC vs. AUMC). The buckets are the following: one bucket for less than one day, one bucket each for days 1 through 7, one bucket for the interval between 7 and 14 days, and one bucket for more than 14 days.

# C  Training Details

In this section, we provide details on the hyperparameters tuning. Table 7 lists the tuning range of all hyperparameters. To avoid repetition, we list hyperparameters that appear at all methods in the first rows of Table 7. For each dataset (benchmark or medical) and each task (i. e., decompensation, mortality, and length of stay prediction), we performed a grid search for hyperparameter tuning *separately* for each method.

We implemented our CLUDA framework and all the baseline methods in PyTorch. For this, we carefully considered the original implementations and the benchmarking suites (Cai et al., 2021; Chen et al., 2020a; Kang et al., 2019; Liu & Xue, 2021; Long et al., 2018; Purushotham et al., 2017; Ragab et al., 2022; Rahman et al., 2020; Sun & Saenko, 2016; Tzeng et al., 2014; Wilson et al., 2020; Zhu et al., 2020). For training and testing, we used NVIDIA GeForce GTX 1080 Ti with 11GB GPU memory. We minimize the loss of each method via Adam optimizer.

We implement the feature extractor $F(\cdot)$ via a temporal convolutional network (TCN) (Bai et al., 2018). We set its kernel size 3 and dilation factor 2. For benchmark datasets, we use 6 layers with 16 channels, whereas for medical datasets, we use 5 layers with 64 channels. This configuration remains the same across all methods so that the difference in prediction performance is attributed to their novel UDA approach.

We now explain how we decide the search range of the hyperparameters (e. g., learning rate, weight decay). The low learning rate is preferred so that the methods converge to a certain loss after seeing all samples from each dataset. Especially, the medical dataset MIMIC has roughly 2.4M samples, and it requires $\sim$1.2K steps to iterate over all these samples with a batch size of 2048. With higher learning rates, the methods converge to a loss even before one iteration over the dataset. We observed that this leads to suboptimal prediction performance (i.e., lower AUROC, AUPRC and KAPPA scores). For the hyperparameters regarding the contrastive learning framework, we are informed by the configuration of MoCo(He et al., 2020) as a starting point. We explored a certain range to improve the performance. For the feature extractor $F(\cdot)$ and the classifier $C(\cdot)$, we used the best hyperparameter configuration obtained by w/o UDA as a starting point. For benchmark datasets, we trained all methods for max. 5,000 training steps with a batch size of 128. For medical datasets, we trained all methods for max. 30,000 training steps with a batch size of 2048 (except AdvSKM, DDC, DSAN, and MMDA with a batch size of 1024 to fit into GPU).

For early stopping and hyperparameter selection, we deliberately avoided the use of data from the labeled target domain. In our work, we aim to present the performance results as close as possible to the real-world scenario of UDA in, e. g., medical practice. We applied early stopping based on the validation loss, which involves labeled source domain and unlabeled target domain (as the overall loss of CLUDA in Sec. 4.5). For hyperparameter selection, we adopted the following two-way approach. For the hyperparameters regarding the model architecture (e. g., num. layers, hidden dimensions, etc.) and the training approach (e. g. learning rate, weight decay etc.) with the fixed weights of the loss components (e. g., $\lambda_{\text{disc}}$, $\lambda_{\text{CL}}$ of our CLUDA or weight MMD loss of DDC), we considered the validation loss as for the early stopping. However, for the hyperparameters regarding the loss components, validation losses are not comparable because the loss values are in different scale. (Trivially, one could disable some of the loss components, i. e., by setting the weights to 0, and hence get a lower validation loss. However, this would not result in a better performance on target domain). Therefore, to select the model across different loss weights, we choose the one with the highest performance metric (e. g., accuracy, macro F1, or AUROC depending on the setting) on the labeled validation source domain as our proxy. This choice is informed by the theory of learning from different domains (Ben-David et al., 2010) in that the loss on the target domain is upper-bounded by the loss on the source domain and some other additional terms. Hence, we are aiming for a better bound on the target domain by choosing a better performance on the source domain. After the model selection, we report the prediction results on the labeled test set from the target domain (and from source domain in Sec. 6.3), yet which have never been seen during training or model selection. We applied the same procedure to all the baseline methods in our the paper to ensure a fair comparison. To report variability in the test performance of each method, we repeated each experiment with 10 different random seeds (i. e., 10 different random initializations) and then show error bars.

Here, we compare the runtimes of each method. For this, we use MIMIC-III (the largest dataset in our experiments). We report average runtimes per 100 training steps since the total runtime (i. e., total

number of training steps) varies with the step of early stopping applied at each run. For each method, the average runtimes (per 100 training steps) are the following: 44.83 seconds for **w/o UDA**, 122.81 seconds for **VRADA**, 81.06 seconds for **CoDATS**, 151.20 seconds for **TS-SASA**, 73.67 seconds for **AdvSKM** *with a half batch size*, 119.42 seconds for **CAN**, 83.93 seconds for **CDAN**, 59.92 seconds for **DDC** *with a half batch size*, 85.67 seconds for **DeepCORAL**, 62.38 seconds for **DSAN** *with a half batch size*, 83.81 seconds for **HoMM**, 68.92 seconds for **MMDA** *with a half batch size*, and 96.11 seconds for our **CLUDA**.

### AUGMENTATIONS

To capture the contextual representation of medical time series, we apply semantic-preserving augmentations (Cheng et al., 2020; Kiyasseh et al., 2021; Yèche et al., 2021) in our CLUDA framework. We list the augmentations and their optimal hyperparameters (search range in parenthesis) below:

**History crop:** We mask out a minimum 20 % (10 % – 40 %) of the initial time series with 50 % (20 % – 50 %) probability.

**History cutout:** We mask out a random 15 % time-window (5 % – 20 %) of time series with 50 % (20 % – 70 %) probability.

**Channel dropout:** We mask out each channel (i. e., type of measurement) independently with 10 % (5 % – 30 %) probability.

**Gaussian noise:** We apply Gaussian noise to each measurement independently with standard deviation of 0.1 (0.05 – 0.2).

We apply these augmentations sequentially to each time series twice. As a result, we have two semantic-preserving augmented views of the same time series for our CLUDA framework. Of note, we trained all the baseline methods with and without the augmentations of time series. We always report the their best results.

Table 7: Hyperparameter tuning.

| Method | Hyperparameter | Tuning Range |
|---|---|---|
| All methods, w/o UDA | Classifier hidden dim. | 64, 128, 256 |
| | Batch normalization | True, False |
| | Dropout | 0, 0.1, 0.2 |
| | Learning rate | $5 \cdot 10^{-5}, 2 \cdot 10^{-4}, 5 \cdot 10^{-4}$ |
| | Weight decay | $1 \cdot 10^{-4}, 1 \cdot 10^{-3}, 1 \cdot 10^{-4}$ |
| VRADAPurushotham et al. (2017) | VRNN hidden dim. | 32, 62, 128 |
| | VRNN latent dim. | 32, 64, 128 |
| | VRNN num. layers | 1, 2, 3 |
| | Discriminator hidden dim. | 64, 128, 256 |
| | Weight discriminator loss | 0.1, 0.5, 1 |
| | Weight KL divergence | 0.1, 0.5, 1 |
| | Weight neg. log-likelihood | 0.1, 0.5, 1 |
| CoDATSWilson et al. (2020) | Discriminator hidden dim. | 64, 128, 256 |
| | Weight discriminator loss | 0.1, 0.5, 1 |
| TS-SASACai et al. (2021) | LSTM hidden dim | 4, 8, 12 |
| | Num. segments | 4, 8, 12, 24 |
| | Segment lengths | 3, 6, 12, 24 |
| | MMD kernel type | Linear, Gaussian |
| | Weight intra-attention loss | 0.1, 0.5, 1 |
| | Weight inter-attention loss | 0.1, 0.5, 1 |
| AdvSKMLiu & Xue (2021) | Spectral kernel hidden dim. | 32, 64, 128 |
| | Spectral kernel output dim. | 32, 64, 128 |
| | Spectral kernel type | Linear, Gaussian |
| | Num. kernel (if Gaussian) | 3, 5, 7 |
| | Weight MMD loss | 0.1, 0.5, 1 |
| CANKang et al. (2019) | Kernel type | Linear, Gaussian |
| | Num. kernel (if Gaussian) | 1, 3, 5, 7 |
| | Num. iterations k-means clustering (each loop) | 1,3,5 |
| | Sampling type | Random, Class-aware |
| | Weight MMD loss | 0.1, 0.5, 1 |
| CDANLong et al. (2018) | Discriminator hidden dim. | 64, 128, 256 |
| | Multiplier discriminator update | 0.1, 1, 10 |
| | Weight discriminator loss | 0.1, 0.5, 1 |
| | Weight conditional entropy loss | 0.1, 0.5, 1 |
| DDCTzeng et al. (2014) | Kernel type | Linear, Gaussian |
| | Num. kernel (if Gaussian) | 1, 3, 5, 7 |
| | Weight MMD loss | 0.1, 0.5, 1 |
| DeepCORALSun & Saenko (2016) | Weight CORAL loss | 0.1, 0.3, 0.5, 1 |
| DSANZhu et al. (2020) | Kernel multiplier | 1, 2, 3 |
| | Num. kernel | 3, 5, 7 |
| | Weight domain loss | 0.1, 0.5, 1 |
| HoMMChen et al. (2020a) | Moment order | 1, 2, 3 |
| | Weight domain discrepancy loss | 0.1, 0.5, 1 |
| | Weight discriminative clustering loss | 0.1, 0.5, 1 |
| MMDARahman et al. (2020) | Kernel type | Linear, Gaussian |
| | Num. kernel (if Gaussian) | 1, 3, 5, 7 |
| | Weight MMD loss | 0.1, 0.5, 1 |
| | Weight CORAL loss | 0.1, 0.5, 1 |
| | Weight Entropy loss | 0.1, 0.5, 1 |
| CLUDA (ours) | Momentum | 0.9, 0.95, 0.99 |
| | Queue size | 24576, 49152, 98304 |
| | Discriminator hidden dim. | 64, 128, 256 |
| | Projector hidden dim. | 64, 128, 256 |
| | $\lambda_{\text{disc}}$ | 0.1, 0.5, 1 |
| | $\lambda_{\text{CL}}$ | 0.05, 0.1, 0.2 |
| | $\lambda_{\text{NNCL}}$ | 0.05, 0.1, 0.2 |

# D   UDA ON BENCHMARK DATASETS

We perform the activity prediction as a UDA task based on the benchmark datasets WISDM, HAR, and HHAR. For each dataset, we present the prediction results for 10 randomly selected source-target pairs. For each source-target pair, we repeat the experiments with 10 random initializations and report the mean values. Table 8 shows the accuracy on the target domains and average accuracy for each dataset. Similarly, Table 9 shows the Macro-F1 on the target domains and average Macro-F1 for each dataset.

Table 8: Activity prediction for each dataset between various subjects. Shown: mean Accuracy over 10 random initializations.

| Sour ↦ Tar | w/o UDA | VRADA | CoDATS | AdvSKM | CAN | CDAN | DDC | DeepCORAL | DSAN | HoMM | MMDA | CLUDA (ours) |
|---|---|---|---|---|---|---|---|---|---|---|---|---|
| WISDM 12 ↦ 19 | **0.745** | 0.558 | 0.633 | 0.639 | 0.594 | 0.488 | 0.564 | 0.433 | 0.639 | 0.415 | 0.358 | 0.694 |
| WISDM 12 ↦ 7 | 0.654 | 0.708 | 0.721 | 0.742 | 0.588 | 0.771 | 0.692 | 0.592 | 0.625 | 0.546 | 0.679 | **0.792** |
| WISDM 18 ↦ 20 | 0.385 | 0.571 | 0.634 | 0.390 | 0.439 | 0.771 | 0.390 | 0.380 | 0.366 | 0.429 | 0.380 | **0.780** |
| WISDM 19 ↦ 2 | 0.410 | **0.644** | 0.395 | 0.434 | 0.322 | 0.346 | 0.459 | 0.473 | 0.366 | 0.488 | 0.385 | 0.561 |
| WISDM 2 ↦ 28 | 0.787 | 0.729 | 0.809 | 0.809 | 0.760 | 0.813 | 0.782 | 0.827 | 0.773 | 0.787 | 0.813 | **0.849** |
| WISDM 26 ↦ 2 | 0.634 | 0.683 | 0.727 | 0.620 | 0.580 | 0.615 | 0.600 | 0.737 | 0.605 | 0.702 | 0.634 | **0.863** |
| WISDM 28 ↦ 2 | 0.702 | 0.688 | 0.717 | 0.707 | 0.561 | 0.580 | 0.702 | 0.649 | 0.673 | 0.644 | 0.668 | **0.741** |
| WISDM 28 ↦ 20 | 0.727 | 0.741 | 0.741 | 0.707 | 0.673 | 0.776 | 0.727 | 0.737 | 0.746 | 0.790 | 0.722 | **0.820** |
| WISDM 7 ↦ 2 | 0.620 | 0.605 | 0.610 | 0.610 | 0.571 | 0.649 | 0.620 | 0.624 | 0.620 | 0.605 | 0.605 | **0.712** |
| WISDM 7 ↦ 26 | 0.722 | 0.693 | 0.702 | 0.702 | 0.717 | 0.722 | 0.717 | 0.683 | 0.698 | 0.698 | 0.712 | **0.727** |
| WISDM Avg | 0.639 | 0.662 | 0.669 | 0.636 | 0.580 | 0.653 | 0.625 | 0.613 | 0.611 | 0.610 | 0.596 | **0.754** |
| HAR 15 ↦ 19 | 0.722 | 0.756 | 0.733 | 0.741 | 0.685 | 0.759 | 0.733 | 0.759 | 0.874 | 0.748 | 0.726 | **0.967** |
| HAR 18 ↦ 21 | 0.552 | 0.794 | 0.552 | 0.555 | 0.552 | 0.803 | 0.548 | 0.610 | 0.558 | 0.581 | 0.555 | **0.910** |
| HAR 19 ↦ 25 | 0.461 | 0.768 | 0.468 | 0.452 | 0.661 | 0.771 | 0.455 | 0.590 | 0.774 | 0.487 | 0.448 | **0.932** |
| HAR 19 ↦ 27 | 0.751 | 0.793 | 0.709 | 0.723 | 0.782 | 0.807 | 0.747 | 0.744 | 0.891 | 0.726 | 0.754 | **0.996** |
| HAR 20 ↦ 6 | 0.616 | 0.808 | 0.661 | 0.641 | 0.747 | 0.820 | 0.608 | 0.686 | 0.784 | 0.673 | 0.694 | **1.000** |
| HAR 23 ↦ 13 | 0.448 | 0.736 | 0.504 | 0.504 | 0.476 | 0.700 | 0.504 | 0.668 | 0.628 | 0.604 | 0.572 | **0.788** |
| HAR 24 ↦ 22 | 0.808 | 0.837 | 0.820 | 0.833 | 0.820 | 0.837 | 0.808 | 0.743 | 0.808 | 0.853 | 0.829 | **0.988** |
| HAR 25 ↦ 24 | 0.545 | 0.817 | 0.583 | 0.566 | 0.721 | 0.790 | 0.593 | 0.648 | 0.883 | 0.607 | 0.666 | **0.993** |
| HAR 3 ↦ 20 | 0.852 | 0.752 | 0.874 | 0.878 | 0.652 | 0.815 | 0.885 | 0.848 | 0.804 | 0.874 | 0.815 | **0.967** |
| HAR 13 ↦ 19 | 0.796 | 0.752 | 0.793 | 0.807 | 0.785 | 0.841 | 0.800 | 0.793 | 0.726 | 0.815 | 0.800 | **0.904** |
| HAR Avg | 0.655 | 0.781 | 0.670 | 0.670 | 0.688 | 0.794 | 0.668 | 0.709 | 0.773 | 0.697 | 0.686 | **0.944** |
| HHAR 0 ↦ 2 | 0.656 | 0.593 | 0.650 | 0.681 | 0.721 | 0.676 | 0.659 | 0.618 | 0.292 | 0.680 | 0.671 | **0.726** |
| HHAR 1 ↦ 6 | 0.673 | 0.690 | 0.686 | 0.652 | 0.619 | 0.717 | 0.672 | 0.712 | 0.689 | 0.725 | 0.686 | **0.855** |
| HHAR 2 ↦ 4 | 0.296 | 0.476 | 0.381 | 0.291 | 0.391 | 0.472 | 0.304 | 0.332 | 0.229 | 0.332 | 0.238 | **0.585** |
| HHAR 4 ↦ 0 | 0.183 | 0.263 | 0.229 | 0.203 | 0.194 | 0.262 | 0.216 | 0.259 | 0.193 | 0.193 | 0.205 | **0.353** |
| HHAR 4 ↦ 1 | 0.454 | 0.558 | 0.501 | 0.494 | 0.549 | 0.690 | 0.502 | 0.482 | 0.504 | 0.628 | 0.551 | **0.774** |
| HHAR 5 ↦ 1 | 0.757 | 0.775 | 0.761 | 0.737 | 0.829 | 0.857 | 0.744 | 0.787 | 0.407 | 0.784 | 0.790 | **0.948** |
| HHAR 7 ↦ 1 | 0.358 | 0.575 | 0.551 | 0.426 | 0.534 | 0.413 | 0.378 | 0.511 | 0.366 | 0.496 | 0.415 | **0.875** |
| HHAR 7 ↦ 5 | 0.199 | 0.523 | 0.380 | 0.192 | 0.592 | 0.492 | 0.229 | 0.489 | 0.233 | 0.328 | 0.320 | **0.636** |
| HHAR 8 ↦ 3 | 0.760 | 0.813 | 0.766 | 0.748 | 0.860 | **0.942** | 0.763 | 0.869 | 0.602 | 0.844 | 0.934 | **0.942** |
| HHAR 8 ↦ 4 | 0.627 | 0.720 | 0.601 | 0.650 | 0.660 | 0.712 | 0.629 | 0.618 | 0.516 | 0.658 | 0.701 | **0.896** |
| HHAR Avg | 0.496 | 0.599 | 0.551 | 0.508 | 0.595 | 0.623 | 0.510 | 0.568 | 0.403 | 0.567 | 0.551 | **0.759** |

Higher is better. Best value in bold.

Overall, our CLUDA outperforms the UDA baselines by a large margin, as discussed in the main paper. Specifically, CLUDA achieves the best accuracy in 28 out of 30 UDA scenarios and the best Macro-F1 in 27 out of 30 UDA scenarios. Thereby, the results confirm the effectiveness of our method.

Table 9: Activity prediction for each dataset between various subjects. Shown: mean MacroF1 over 10 random initializations.

| Sour ↦ Tar | w/o UDA | VRADA | CoDATS | AdvSKM | CAN | CDAN | DDC | DeepCORAL | DSAN | HoMM | MMDA | CLUDA (ours) |
|---|---|---|---|---|---|---|---|---|---|---|---|---|
| WISDM 12 ↦ 19 | **0.577** | 0.410 | 0.456 | 0.510 | 0.508 | 0.298 | 0.396 | 0.317 | 0.518 | 0.281 | 0.233 | 0.532 |
| WISDM 12 ↦ 7 | 0.543 | 0.437 | 0.612 | 0.655 | 0.636 | 0.546 | 0.632 | 0.486 | 0.574 | 0.442 | 0.539 | **0.678** |
| WISDM 18 ↦ 20 | 0.339 | 0.578 | 0.427 | 0.348 | 0.389 | 0.600 | 0.383 | 0.379 | 0.268 | 0.421 | 0.280 | **0.673** |
| WISDM 19 ↦ 2 | 0.436 | **0.615** | 0.403 | 0.460 | 0.327 | 0.312 | 0.459 | 0.501 | 0.428 | 0.522 | 0.306 | 0.458 |
| WISDM 2 ↦ 28 | 0.696 | 0.688 | 0.688 | 0.742 | 0.610 | 0.644 | 0.669 | 0.726 | 0.654 | 0.691 | 0.677 | **0.788** |
| WISDM 26 ↦ 2 | 0.472 | 0.517 | 0.598 | 0.463 | 0.362 | 0.404 | 0.414 | 0.618 | 0.424 | 0.519 | 0.453 | **0.701** |
| WISDM 28 ↦ 2 | 0.450 | 0.473 | 0.492 | 0.484 | 0.412 | 0.400 | 0.484 | 0.495 | 0.451 | 0.511 | 0.430 | **0.710** |
| WISDM 28 ↦ 20 | 0.560 | 0.672 | 0.578 | 0.557 | 0.655 | 0.605 | 0.571 | 0.620 | 0.615 | 0.699 | 0.537 | **0.703** |
| WISDM 7 ↦ 2 | 0.443 | 0.399 | 0.494 | 0.476 | 0.490 | 0.543 | 0.496 | 0.490 | 0.481 | 0.494 | 0.459 | **0.576** |
| WISDM 7 ↦ 26 | 0.407 | 0.308 | 0.405 | **0.416** | 0.395 | 0.344 | 0.412 | 0.396 | 0.401 | 0.406 | 0.385 | 0.403 |
| WISDM Avg | 0.492 | 0.510 | 0.515 | 0.511 | 0.479 | 0.469 | 0.492 | 0.503 | 0.482 | 0.498 | 0.430 | **0.622** |
| HAR 15 ↦ 19 | 0.647 | 0.657 | 0.663 | 0.664 | 0.593 | 0.696 | 0.658 | 0.708 | 0.831 | 0.686 | 0.656 | **0.957** |
| HAR 18 ↦ 21 | 0.431 | 0.668 | 0.428 | 0.445 | 0.434 | 0.718 | 0.427 | 0.539 | 0.458 | 0.486 | 0.440 | **0.923** |
| HAR 19 ↦ 25 | 0.369 | 0.737 | 0.381 | 0.359 | 0.640 | 0.768 | 0.360 | 0.535 | 0.754 | 0.397 | 0.348 | **0.932** |
| HAR 19 ↦ 27 | 0.685 | 0.723 | 0.643 | 0.652 | 0.723 | 0.752 | 0.683 | 0.689 | 0.852 | 0.650 | 0.684 | **0.996** |
| HAR 20 ↦ 6 | 0.539 | 0.773 | 0.603 | 0.576 | 0.725 | 0.796 | 0.529 | 0.666 | 0.759 | 0.627 | 0.641 | **1.000** |
| HAR 23 ↦ 13 | 0.377 | 0.696 | 0.440 | 0.436 | 0.410 | 0.660 | 0.447 | 0.616 | 0.606 | 0.549 | 0.527 | **0.762** |
| HAR 24 ↦ 22 | 0.712 | 0.749 | 0.714 | 0.726 | 0.772 | 0.756 | 0.710 | 0.647 | 0.726 | 0.768 | 0.722 | **0.983** |
| HAR 25 ↦ 24 | 0.488 | 0.782 | 0.516 | 0.503 | 0.702 | 0.765 | 0.527 | 0.625 | 0.873 | 0.538 | 0.641 | **0.992** |
| HAR 3 ↦ 20 | 0.813 | 0.671 | 0.853 | 0.847 | 0.549 | 0.769 | 0.852 | 0.828 | 0.757 | 0.860 | 0.784 | **0.968** |
| HAR 13 ↦ 19 | 0.743 | 0.696 | 0.738 | 0.769 | 0.729 | 0.837 | 0.752 | 0.763 | 0.662 | 0.798 | 0.752 | **0.911** |
| HAR Avg | 0.580 | 0.715 | 0.598 | 0.598 | 0.628 | 0.752 | 0.595 | 0.662 | 0.728 | 0.636 | 0.619 | **0.942** |
| HHAR 0 ↦ 2 | 0.606 | 0.536 | 0.598 | 0.628 | 0.667 | 0.611 | 0.605 | 0.569 | 0.205 | 0.627 | 0.612 | **0.710** |
| HHAR 1 ↦ 6 | 0.685 | 0.702 | 0.696 | 0.662 | 0.621 | 0.727 | 0.678 | 0.725 | 0.696 | 0.726 | 0.693 | **0.858** |
| HHAR 2 ↦ 4 | 0.196 | 0.415 | 0.320 | 0.219 | 0.294 | 0.431 | 0.231 | 0.305 | 0.143 | 0.230 | 0.192 | **0.526** |
| HHAR 4 ↦ 0 | 0.147 | 0.243 | 0.222 | 0.163 | 0.165 | 0.273 | 0.175 | 0.249 | 0.116 | 0.179 | 0.162 | **0.352** |
| HHAR 4 ↦ 1 | 0.415 | 0.545 | 0.469 | 0.466 | 0.523 | 0.667 | 0.456 | 0.461 | 0.488 | 0.607 | 0.517 | **0.751** |
| HHAR 5 ↦ 1 | 0.711 | 0.756 | 0.723 | 0.692 | 0.813 | 0.848 | 0.707 | 0.766 | 0.285 | 0.738 | 0.765 | **0.950** |
| HHAR 7 ↦ 1 | 0.275 | 0.583 | 0.528 | 0.338 | 0.524 | 0.412 | 0.280 | 0.483 | 0.278 | 0.461 | 0.367 | **0.875** |
| HHAR 7 ↦ 5 | 0.151 | 0.529 | 0.374 | 0.154 | 0.546 | 0.480 | 0.175 | 0.496 | 0.192 | 0.323 | 0.283 | **0.626** |
| HHAR 8 ↦ 3 | 0.701 | 0.818 | 0.734 | 0.692 | 0.845 | 0.943 | 0.719 | 0.872 | 0.564 | 0.836 | 0.936 | **0.944** |
| HHAR 8 ↦ 4 | 0.542 | 0.715 | 0.539 | 0.580 | 0.596 | 0.710 | 0.550 | 0.578 | 0.434 | 0.606 | 0.636 | **0.891** |
| HHAR Avg | 0.443 | 0.584 | 0.520 | 0.459 | 0.559 | 0.610 | 0.458 | 0.550 | 0.340 | 0.533 | 0.516 | **0.748** |

Higher is better. Best value in bold.

# E  EMBEDDING VISUALIZATION

In this section, we provide the t-SNE visualization (see Fig. 10) for the embeddings of each method from HHAR dataset. When there is no domain adaptation (see Fig. 10a of w/o UDA), there is a significant domain shift between source and target. As a result, embeddings of one class in target domain overlap with embeddings of another class in source domain. Thereby, the classifier learned on the source domain cannot generalize well over the target domain. The UDA baselines mitigate the domain shift; however, they still mix several classes. On the other hand, our CLUDA clearly pulls the embeddings of the same class (even though they are in different domains), and facilitates better generalization in the target domain.

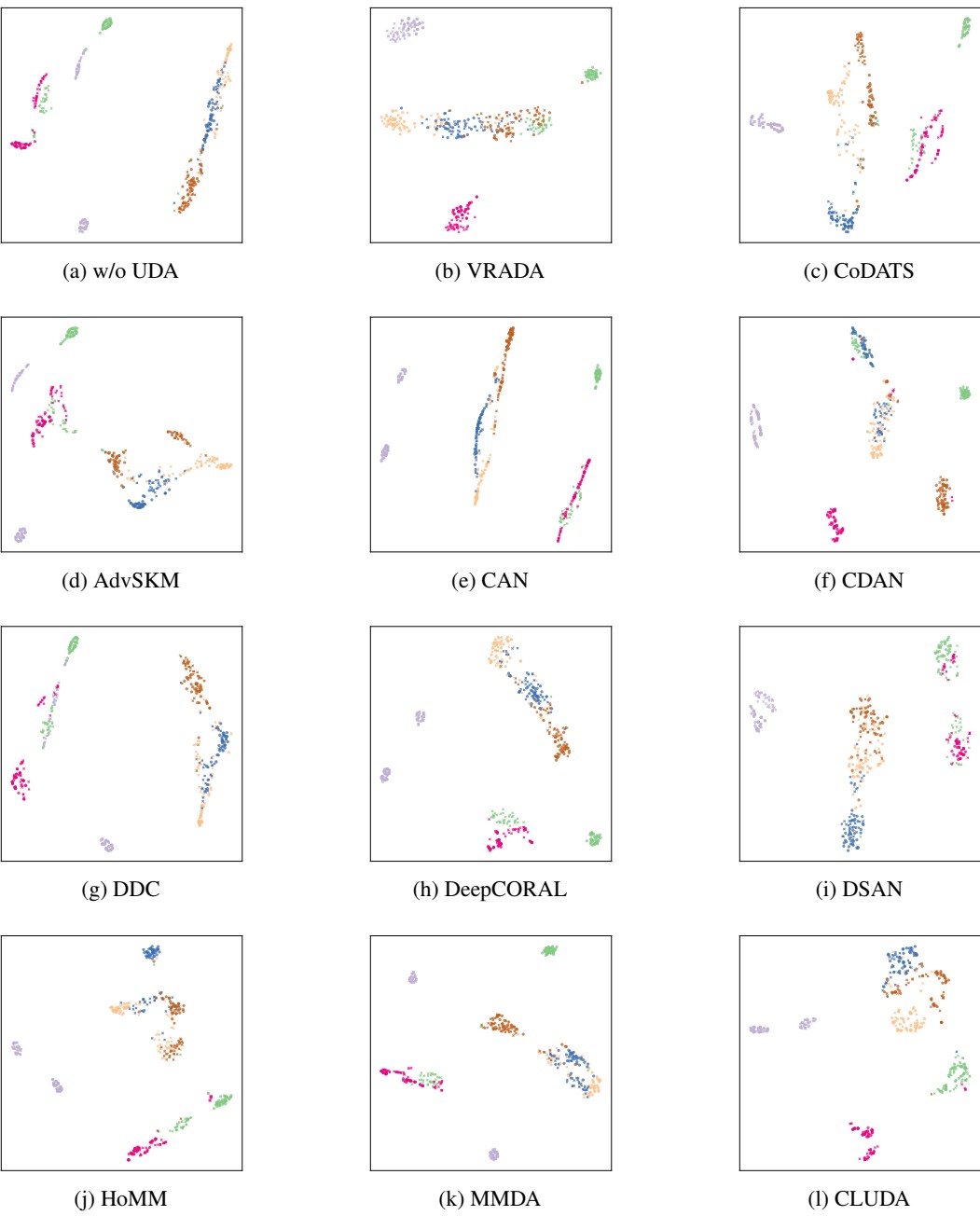

Figure 10: t-SNE visualization of the embeddings from each model on HHAR dataset. Each class is represented by a different color. Shape shows source and target domains (circle vs. cross).

# F    Ablation Study for UDA on Benchmark Datasets

We further conduct an ablation study on the benchmark datasets WISDM, HAR, and HHAR. We use the same variants of CLUDA from the main paper (see Sec. 6.1: **w/o CL and w/o NNCL**, **w/o CL**, **w/o NNCL**, and **w/o Discriminator**. Similar to the main experiments, for each dataset, we present the prediction results for 10 randomly selected source-target pairs. For each source-target pair, we repeat the experiments with 10 random initializations and report the mean values. Table 10 shows the accuracy on the target domains and average accuracy for each dataset. Similarly, Table 11 shows the Macro-F1 on the target domains and average Macro-F1 for each dataset.

Table 10: Activity prediction for each dataset between various subjects. Shown: mean Accuracy over 10 random initializations.

| Sour ↦ Tar | w/o UDA | w/o CL and w/o NNCL | w/o CL | w/o NNCL | w/o Discriminator | CLUDA (ours) |
|---|---|---|---|---|---|---|
| WISDM 12 ↦ 19 | 0.745 | 0.433 | 0.770 | 0.470 | **0.803** | 0.694 |
| WISDM 12 ↦ 7 | 0.654 | 0.583 | 0.542 | 0.700 | 0.700 | **0.792** |
| WISDM 18 ↦ 20 | 0.385 | 0.595 | 0.473 | 0.717 | 0.463 | **0.780** |
| WISDM 19 ↦ 2 | 0.410 | 0.410 | 0.463 | 0.517 | 0.527 | **0.561** |
| WISDM 2 ↦ 28 | 0.787 | 0.729 | 0.716 | 0.707 | 0.747 | **0.849** |
| WISDM 26 ↦ 2 | 0.634 | 0.654 | 0.693 | 0.810 | 0.824 | **0.863** |
| WISDM 28 ↦ 2 | 0.702 | 0.337 | 0.507 | 0.522 | **0.761** | 0.741 |
| WISDM 28 ↦ 20 | 0.727 | 0.780 | 0.673 | 0.771 | 0.795 | **0.820** |
| WISDM 7 ↦ 2 | 0.620 | 0.634 | 0.659 | 0.688 | **0.741** | 0.712 |
| WISDM 7 ↦ 26 | 0.722 | 0.707 | 0.707 | 0.678 | 0.707 | **0.727** |
| WISDM Avg | 0.639 | 0.586 | 0.620 | 0.658 | 0.707 | **0.754** |
| HAR 15 ↦ 19 | 0.722 | 0.793 | 0.711 | 0.804 | 0.807 | **0.967** |
| HAR 18 ↦ 21 | 0.552 | 0.813 | 0.806 | 0.855 | 0.861 | **0.910** |
| HAR 19 ↦ 25 | 0.461 | 0.758 | 0.652 | 0.800 | 0.652 | **0.932** |
| HAR 19 ↦ 27 | 0.751 | 0.933 | 0.937 | 0.944 | 0.832 | **0.996** |
| HAR 20 ↦ 6 | 0.616 | 0.959 | 0.910 | 0.865 | 0.906 | **1.000** |
| HAR 23 ↦ 13 | 0.448 | 0.696 | 0.680 | 0.668 | 0.740 | **0.788** |
| HAR 24 ↦ 22 | 0.808 | 0.837 | 0.873 | 0.918 | 0.898 | **0.988** |
| HAR 25 ↦ 24 | 0.545 | 0.938 | 0.890 | 0.928 | 0.910 | **0.993** |
| HAR 3 ↦ 20 | 0.852 | 0.926 | 0.800 | 0.874 | 0.819 | **0.967** |
| HAR 13 ↦ 19 | 0.796 | 0.678 | 0.752 | 0.759 | 0.807 | **0.904** |
| HAR Avg | 0.655 | 0.833 | 0.801 | 0.841 | 0.823 | **0.944** |
| HHAR 0 ↦ 2 | 0.656 | 0.666 | 0.677 | 0.606 | 0.725 | **0.726** |
| HHAR 1 ↦ 6 | 0.673 | 0.735 | 0.731 | 0.771 | 0.790 | **0.855** |
| HHAR 2 ↦ 4 | 0.296 | 0.530 | 0.393 | 0.554 | 0.570 | **0.585** |
| HHAR 4 ↦ 0 | 0.183 | 0.197 | 0.210 | 0.276 | 0.348 | **0.353** |
| HHAR 4 ↦ 1 | 0.454 | 0.536 | 0.711 | 0.554 | **0.782** | 0.774 |
| HHAR 5 ↦ 1 | 0.757 | 0.817 | 0.887 | 0.866 | 0.914 | **0.948** |
| HHAR 7 ↦ 1 | 0.358 | 0.493 | 0.660 | 0.557 | 0.728 | **0.875** |
| HHAR 7 ↦ 5 | 0.199 | 0.357 | 0.460 | 0.423 | 0.547 | **0.636** |
| HHAR 8 ↦ 3 | 0.760 | 0.836 | 0.821 | 0.864 | 0.888 | **0.942** |
| HHAR 8 ↦ 4 | 0.627 | 0.644 | 0.555 | 0.671 | 0.710 | **0.896** |
| HHAR Avg | 0.496 | 0.581 | 0.610 | 0.614 | 0.700 | **0.759** |

Higher is better. Best value in bold.

Overall, our complete CLUDA outperforms all its variants by a significant margin, which confirms our chosen architecture.

Table 11: Activity prediction for each dataset between various subjects. Shown: mean MacroF1 over 10 random initializations.

| Sour ↦ Tar | w/o UDA | w/o CL and w/o NNCL | w/o CL | w/o NNCL | w/o Discriminator | CLUDA (ours) |
|---|---|---|---|---|---|---|
| WISDM 12 ↦ 19 | 0.577 | 0.334 | 0.606 | 0.309 | **0.620** | 0.532 |
| WISDM 12 ↦ 7 | 0.543 | 0.458 | 0.468 | 0.534 | 0.525 | **0.678** |
| WISDM 18 ↦ 20 | 0.339 | 0.485 | 0.481 | 0.507 | 0.523 | **0.673** |
| WISDM 19 ↦ 2 | 0.436 | 0.358 | 0.492 | 0.415 | **0.565** | 0.458 |
| WISDM 2 ↦ 28 | 0.696 | 0.669 | 0.685 | 0.682 | 0.667 | **0.788** |
| WISDM 26 ↦ 2 | 0.472 | 0.367 | 0.494 | 0.620 | 0.642 | **0.701** |
| WISDM 28 ↦ 2 | 0.450 | 0.373 | 0.469 | 0.457 | 0.638 | **0.710** |
| WISDM 28 ↦ 20 | 0.560 | 0.652 | 0.547 | 0.619 | 0.686 | **0.703** |
| WISDM 7 ↦ 2 | 0.443 | 0.418 | 0.455 | 0.499 | 0.556 | **0.576** |
| WISDM 7 ↦ 26 | 0.407 | 0.321 | 0.337 | 0.341 | 0.343 | **0.403** |
| WISDM Avg | 0.492 | 0.444 | 0.503 | 0.498 | 0.577 | **0.622** |
| HAR 15 ↦ 19 | 0.647 | 0.730 | 0.622 | 0.783 | 0.746 | **0.957** |
| HAR 18 ↦ 21 | 0.431 | 0.746 | 0.748 | 0.840 | 0.835 | **0.923** |
| HAR 19 ↦ 25 | 0.369 | 0.755 | 0.564 | 0.797 | 0.603 | **0.932** |
| HAR 19 ↦ 27 | 0.685 | 0.896 | 0.918 | 0.922 | 0.775 | **0.996** |
| HAR 20 ↦ 6 | 0.539 | 0.961 | 0.900 | 0.855 | 0.912 | **1.000** |
| HAR 23 ↦ 13 | 0.377 | 0.670 | 0.607 | 0.638 | 0.687 | **0.762** |
| HAR 24 ↦ 22 | 0.712 | 0.797 | 0.798 | 0.883 | 0.848 | **0.983** |
| HAR 25 ↦ 24 | 0.488 | 0.926 | 0.861 | 0.917 | 0.899 | **0.992** |
| HAR 3 ↦ 20 | 0.813 | 0.920 | 0.713 | 0.835 | 0.740 | **0.968** |
| HAR 13 ↦ 19 | 0.743 | 0.681 | 0.680 | 0.777 | 0.785 | **0.911** |
| HAR Avg | 0.580 | 0.808 | 0.741 | 0.825 | 0.783 | **0.942** |
| HHAR 0 ↦ 2 | 0.606 | 0.599 | 0.611 | 0.549 | 0.661 | **0.710** |
| HHAR 1 ↦ 6 | 0.685 | 0.729 | 0.721 | 0.771 | 0.785 | **0.858** |
| HHAR 2 ↦ 4 | 0.196 | 0.464 | 0.272 | 0.484 | 0.493 | **0.526** |
| HHAR 4 ↦ 0 | 0.147 | 0.166 | 0.188 | 0.274 | 0.331 | **0.352** |
| HHAR 4 ↦ 1 | 0.415 | 0.487 | 0.652 | 0.497 | 0.748 | **0.751** |
| HHAR 5 ↦ 1 | 0.711 | 0.809 | 0.877 | 0.864 | 0.916 | **0.950** |
| HHAR 7 ↦ 1 | 0.275 | 0.467 | 0.626 | 0.536 | 0.732 | **0.875** |
| HHAR 7 ↦ 5 | 0.151 | 0.348 | 0.410 | 0.413 | 0.549 | **0.626** |
| HHAR 8 ↦ 3 | 0.701 | 0.822 | 0.809 | 0.859 | 0.876 | **0.944** |
| HHAR 8 ↦ 4 | 0.542 | 0.610 | 0.479 | 0.628 | 0.671 | **0.891** |
| HHAR Avg | 0.443 | 0.550 | 0.565 | 0.588 | 0.676 | **0.748** |

Higher is better. Best value in bold.

# G  UDA ACROSS VARIOUS AGE GROUPS

Following the earlier works (Purushotham et al., 2017; Cai et al., 2021), we conducted extensive experiments to compare the UDA performance of our CLUDA framework across various age groups. We consider the following groups: (1) Group 1: working-age adult (20 to 45 years old patients); (2) Group 2: old working-age adult (46 to 65 years old patients); (3) Group 3: elderly (66 to 85 years old patients); and (4) Group 4: seniors (85+ years old patients). Therefore, within each dataset (MIMIC and AUMC), we list the results of all combinations of Source ↦ Target for mortality prediction (i. e., Group 1 ↦ Group 2, Group 1 ↦ Group 3, . . ., Group 4 ↦ Group 3). Results are shown in Table 1 (in the main paper) for MIMIC and Table 12 for AUMC.

We further extend the experiments to **across datasets**. That means, we pick the source domain as one age group from one dataset (e. g., Group 1 of MIMIC) and pick the target domain as one age group from the other dataset (e. g., Group 3 of AUMC). We, again, conducted the experiments for all combinations of age groups across the datasets. Results are shown in Table 13 from MIMIC to AUMC and Table 14 from AUMC to MIMIC.

We report the mean over 10 random initialization. For better readability, we omitted the standard deviation. Nevertheless, we highlight performance results in bold when corresponding baselines are outperformed at a significant level.

Table 12: Mortality prediction between various age groups of AUMC. Shown: mean AUROC over 10 random initializations.

| Sour ↦ Tar | w/o UDA | VRADA | CoDATS | AdvSKM | CAN | CDAN | DDC | DeepCORAL | DSAN | HoMM | MMDA | CLUDA (ours) |
|---|---|---|---|---|---|---|---|---|---|---|---|---|
| 1 ↦ 2 | 0.557 | **0.583** | 0.545 | 0.562 | 0.536 | 0.551 | 0.527 | 0.538 | 0.502 | 0.538 | 0.536 | 0.571 |
| 1 ↦ 3 | 0.602 | 0.659 | 0.602 | 0.623 | 0.670 | 0.578 | 0.681 | 0.669 | 0.657 | 0.678 | 0.659 | **0.686** |
| 1 ↦ 4 | 0.683 | 0.732 | 0.672 | 0.702 | 0.727 | 0.677 | 0.738 | 0.740 | 0.747 | 0.739 | 0.742 | **0.749** |
| 2 ↦ 1 | 0.719 | 0.716 | 0.728 | 0.740 | 0.629 | 0.658 | **0.749** | 0.685 | 0.701 | 0.690 | 0.709 | 0.743 |
| 2 ↦ 3 | 0.728 | 0.743 | 0.728 | 0.740 | 0.705 | 0.706 | 0.692 | 0.705 | 0.740 | 0.714 | 0.688 | **0.765** |
| 2 ↦ 4 | 0.795 | 0.783 | 0.798 | **0.800** | 0.772 | 0.742 | 0.775 | 0.780 | **0.800** | 0.784 | 0.791 | 0.795 |
| 3 ↦ 1 | 0.780 | 0.761 | 0.778 | 0.781 | 0.722 | 0.636 | 0.745 | 0.733 | 0.732 | 0.745 | 0.726 | **0.812** |
| 3 ↦ 2 | 0.595 | 0.618 | 0.588 | 0.604 | 0.575 | 0.607 | 0.572 | 0.576 | 0.585 | 0.553 | 0.550 | **0.657** |
| 3 ↦ 4 | 0.817 | 0.788 | 0.815 | 0.832 | 0.823 | 0.790 | 0.809 | 0.813 | 0.817 | 0.794 | 0.803 | **0.836** |
| 4 ↦ 1 | 0.730 | 0.739 | 0.727 | 0.738 | 0.718 | 0.671 | 0.726 | 0.690 | 0.683 | 0.721 | **0.740** | 0.731 |
| 4 ↦ 2 | 0.640 | 0.618 | 0.640 | 0.628 | 0.563 | **0.669** | 0.641 | 0.583 | 0.654 | 0.637 | 0.502 | 0.635 |
| 4 ↦ 3 | 0.709 | 0.715 | 0.708 | 0.717 | 0.720 | 0.707 | 0.689 | 0.674 | 0.705 | 0.721 | 0.677 | **0.740** |
| Avg | 0.696 | 0.705 | 0.694 | 0.706 | 0.680 | 0.666 | 0.695 | 0.682 | 0.694 | 0.693 | 0.677 | **0.727** |

Higher is better. Best value in bold. Second best results are underlined if stds overlap.

Table 13: Mortality prediction between various age groups from MIMIC to AUMC. Shown: mean AUROC over 10 random initializations.

| Sour ↦ Tar | w/o UDA | VRADA | CoDATS | AdvSKM | CAN | CDAN | DDC | DeepCORAL | DSAN | HoMM | MMDA | CLUDA (ours) |
|---|---|---|---|---|---|---|---|---|---|---|---|---|
| 1 ↦ 1 | 0.736 | 0.751 | 0.731 | 0.734 | 0.723 | 0.754 | 0.731 | 0.742 | 0.720 | 0.732 | 0.729 | **0.782** |
| 1 ↦ 2 | 0.628 | 0.721 | 0.627 | 0.637 | 0.689 | 0.614 | 0.607 | 0.636 | 0.604 | 0.587 | 0.611 | **0.731** |
| 1 ↦ 3 | 0.662 | 0.688 | 0.657 | 0.671 | 0.656 | 0.654 | 0.618 | 0.661 | 0.629 | 0.630 | 0.653 | **0.707** |
| 1 ↦ 4 | **0.754** | 0.745 | 0.753 | 0.753 | 0.725 | 0.745 | 0.713 | 0.705 | 0.711 | 0.716 | 0.699 | **0.754** |
| 2 ↦ 1 | **0.835** | 0.760 | 0.828 | 0.832 | 0.810 | 0.783 | 0.832 | 0.826 | 0.815 | 0.830 | 0.825 | 0.822 |
| 2 ↦ 2 | 0.629 | 0.699 | 0.633 | 0.635 | 0.634 | 0.691 | 0.631 | 0.630 | 0.634 | 0.633 | 0.636 | **0.705** |
| 2 ↦ 3 | 0.656 | 0.701 | 0.667 | 0.689 | 0.688 | 0.655 | 0.652 | 0.659 | 0.676 | 0.653 | 0.655 | **0.714** |
| 2 ↦ 4 | 0.773 | 0.764 | 0.776 | 0.777 | 0.761 | 0.755 | 0.771 | 0.771 | 0.772 | 0.768 | 0.757 | **0.807** |
| 3 ↦ 1 | 0.763 | 0.748 | 0.754 | 0.776 | 0.762 | 0.746 | 0.764 | 0.769 | 0.729 | 0.753 | 0.737 | **0.789** |
| 3 ↦ 2 | 0.627 | 0.622 | 0.615 | 0.621 | **0.696** | 0.672 | 0.618 | 0.631 | 0.634 | 0.619 | 0.635 | 0.691 |
| 3 ↦ 3 | 0.711 | 0.701 | 0.712 | 0.716 | 0.706 | 0.702 | 0.708 | 0.712 | 0.717 | 0.706 | 0.719 | **0.751** |
| 3 ↦ 4 | 0.782 | 0.750 | 0.784 | 0.785 | 0.772 | 0.756 | 0.784 | 0.786 | 0.785 | 0.788 | 0.771 | **0.796** |
| 4 ↦ 1 | 0.714 | 0.676 | 0.697 | **0.716** | 0.672 | 0.689 | 0.708 | 0.707 | 0.631 | 0.700 | 0.617 | 0.689 |
| 4 ↦ 2 | 0.668 | 0.666 | 0.661 | 0.649 | 0.626 | **0.713** | 0.670 | 0.643 | 0.684 | 0.660 | 0.658 | 0.673 |
| 4 ↦ 3 | 0.619 | 0.627 | 0.614 | 0.606 | 0.617 | 0.620 | 0.616 | 0.609 | 0.626 | 0.613 | 0.610 | **0.635** |
| 4 ↦ 4 | 0.758 | 0.709 | 0.757 | 0.744 | 0.752 | 0.748 | 0.762 | 0.738 | 0.760 | 0.753 | 0.738 | **0.768** |
| Avg | 0.707 | 0.708 | 0.704 | 0.709 | 0.706 | 0.706 | 0.699 | 0.702 | 0.695 | 0.696 | 0.691 | **0.738** |

Higher is better. Best value in bold. Second best results are underlined if stds overlap.

Table 14: Mortality prediction between various age groups from AUMC to MIMIC. Shown: mean AUROC over 10 random initializations.

| Sour ↦ Tar | w/o UDA | VRADA | CoDATS | AdvSKM | CAN | CDAN | DDC | DeepCORAL | DSAN | HoMM | MMDA | CLUDA (ours) |
|---|---|---|---|---|---|---|---|---|---|---|---|---|
| 1 ↦ 1 | 0.693 | 0.733 | 0.694 | 0.698 | 0.738 | 0.681 | 0.713 | 0.714 | 0.722 | 0.714 | 0.708 | **0.791** |
| 1 ↦ 2 | 0.665 | 0.722 | 0.666 | 0.696 | 0.751 | 0.648 | 0.746 | 0.751 | 0.736 | 0.756 | 0.745 | **0.776** |
| 1 ↦ 3 | 0.609 | 0.644 | 0.609 | 0.625 | 0.630 | 0.594 | 0.620 | 0.623 | 0.623 | 0.629 | 0.619 | **0.679** |
| 1 ↦ 4 | 0.600 | 0.579 | 0.599 | **0.609** | 0.584 | 0.585 | 0.584 | 0.593 | 0.603 | 0.590 | 0.551 | 0.598 |
| 2 ↦ 1 | 0.703 | 0.747 | 0.735 | 0.727 | 0.776 | 0.736 | 0.640 | **0.791** | 0.699 | 0.697 | 0.749 | 0.780 |
| 2 ↦ 2 | 0.684 | 0.758 | 0.697 | 0.730 | 0.755 | 0.757 | 0.626 | 0.706 | 0.742 | 0.695 | 0.750 | **0.771** |
| 2 ↦ 3 | 0.641 | 0.693 | 0.648 | 0.659 | 0.664 | 0.677 | 0.625 | 0.675 | 0.676 | 0.645 | 0.637 | **0.702** |
| 2 ↦ 4 | 0.592 | 0.590 | 0.597 | 0.573 | 0.516 | 0.556 | 0.591 | 0.570 | 0.585 | 0.578 | 0.572 | **0.608** |
| 3 ↦ 1 | **0.805** | 0.784 | 0.794 | 0.801 | 0.796 | 0.778 | 0.776 | 0.794 | 0.738 | 0.768 | 0.802 | 0.785 |
| 3 ↦ 2 | 0.751 | 0.769 | 0.747 | 0.747 | 0.732 | 0.698 | 0.738 | 0.746 | 0.683 | 0.744 | 0.743 | **0.774** |
| 3 ↦ 3 | 0.720 | 0.723 | 0.718 | 0.722 | 0.686 | 0.714 | 0.699 | 0.679 | 0.677 | 0.695 | 0.692 | **0.729** |
| 3 ↦ 4 | 0.622 | 0.608 | 0.615 | **0.624** | 0.568 | 0.598 | 0.623 | 0.599 | 0.622 | 0.618 | 0.604 | 0.615 |
| 4 ↦ 1 | 0.801 | 0.756 | 0.808 | 0.819 | 0.796 | 0.786 | 0.709 | **0.831** | 0.701 | 0.806 | 0.734 | 0.819 |
| 4 ↦ 2 | 0.750 | 0.739 | 0.756 | 0.761 | 0.757 | 0.744 | 0.695 | **0.769** | 0.757 | 0.752 | 0.764 | 0.752 |
| 4 ↦ 3 | 0.709 | 0.695 | 0.710 | 0.711 | 0.674 | 0.697 | 0.663 | **0.719** | 0.671 | 0.713 | 0.647 | 0.711 |
| 4 ↦ 4 | 0.697 | 0.664 | 0.695 | **0.698** | 0.645 | 0.684 | 0.663 | 0.641 | 0.684 | 0.693 | 0.643 | 0.679 |
| Avg | 0.690 | 0.700 | 0.693 | 0.700 | 0.692 | 0.683 | 0.669 | 0.700 | 0.682 | 0.693 | 0.685 | **0.723** |

Higher is better. Best value in bold. Second best results are underlined if stds overlap.

Overall, in this section we present 56 prediction tasks to compare the methods across various age groups in both datasets. Out of 56 tasks, our CLUDA achieves the best performance in 36 of them, where it significantly outperforms the other methods. In comparison, the best baseline methods, AdvSKM and DeepCORAL, achieve the best result in only 5 out of 56 tasks. This highlights the consistent and significant performance improvements achieved by our CLUDA in various domains.

# H    ABLATION STUDY FOR UDA ACROSS VARIOUS AGE GROUPS

We further conduct an ablation study to compare different variants of our CLUDA framework. Here, we build upon the previous experiments of various age groups. We use the same variants of CLUDA from the main paper (see Sec. 6.1): **w/o CL and w/o NNCL**, **w/o CL**, **w/o NNCL**, and **w/o Discriminator**. We repeat the results of **w/o UDA** and our CLUDA for better comparability. Table 15 and Table 16 list the UDA performance across age groups within MIMIC and AUMC, respectively. In addition, Tables 17 and 18 list the UDA performance across age groups from MIMIC to AUMC and from AUMC to MIMIC, respectively.

In total, our ablation study counts 56 new experiments. We report the mean over 10 random initialization. For better readability, we omitted the standard deviation. Nevertheless, we highlight performance results in bold when corresponding baselines are outperformed at a significant level.

We make the following important findings. First, our CLUDA works overall best on the target domain, thereby justifying our chosen architecture. Second, the models **w/o CL** and **w/o NNCL** perform significantly worse than our complete framework, which justifies our choice for incorporating both components. Third, we compare **w/o Discriminator** and our CLUDA. As demonstrated by our results, the discriminator is consistently responsible for better UDA for the target domain consistently. Overall, its performance improvement is significant but the gain is smaller than the other components.

Table 15: Mortality prediction between various age groups of MIMIC. Shown: mean AUROC over 10 random initializations.

| Sour $\mapsto$ Tar | w/o UDA | w/o CL and w/o NNCL | w/o CL | w/o NNCL | w/o Discriminator | CLUDA (ours) |
|---|---|---|---|---|---|---|
| $1 \mapsto 2$ | 0.744 | 0.766 | 0.775 | 0.782 | 0.781 | **0.798** |
| $1 \mapsto 3$ | 0.685 | 0.715 | 0.740 | 0.735 | 0.735 | **0.747** |
| $1 \mapsto 4$ | 0.617 | 0.614 | 0.631 | 0.637 | 0.618 | **0.649** |
| $2 \mapsto 1$ | 0.818 | 0.820 | 0.838 | 0.836 | 0.842 | **0.856** |
| $2 \mapsto 3$ | 0.790 | 0.783 | 0.791 | 0.792 | 0.791 | **0.796** |
| $2 \mapsto 4$ | 0.696 | 0.674 | 0.688 | **0.705** | 0.676 | 0.697 |
| $3 \mapsto 1$ | 0.787 | 0.804 | 0.810 | 0.812 | 0.815 | **0.822** |
| $3 \mapsto 2$ | 0.833 | **0.845** | 0.838 | 0.844 | 0.840 | 0.843 |
| $3 \mapsto 4$ | **0.751** | 0.738 | 0.743 | 0.741 | 0.740 | 0.745 |
| $4 \mapsto 1$ | 0.783 | 0.779 | 0.791 | 0.782 | 0.784 | **0.807** |
| $4 \mapsto 2$ | 0.761 | 0.763 | 0.765 | 0.764 | 0.765 | **0.769** |
| $4 \mapsto 3$ | 0.736 | 0.742 | 0.744 | 0.738 | 0.743 | **0.748** |
| Avg | 0.750 | 0.754 | 0.763 | 0.764 | 0.761 | **0.773** |

Higher is better. Best value in bold.

Table 16: Mortality prediction between various age groups of AUMC. Shown: mean AUROC over 10 random initializations.

| Sour $\mapsto$ Tar | w/o UDA | w/o CL and w/o NNCL | w/o CL | w/o NNCL | w/o Discriminator | CLUDA (ours) |
|---|---|---|---|---|---|---|
| $1 \mapsto 2$ | 0.557 | 0.561 | 0.563 | 0.562 | 0.563 | **0.571** |
| $1 \mapsto 3$ | 0.602 | 0.629 | 0.636 | 0.641 | 0.643 | **0.686** |
| $1 \mapsto 4$ | 0.683 | 0.708 | 0.713 | 0.716 | 0.696 | **0.749** |
| $2 \mapsto 1$ | 0.719 | 0.709 | 0.726 | 0.733 | 0.735 | **0.743** |
| $2 \mapsto 3$ | 0.728 | 0.725 | 0.740 | 0.743 | 0.761 | **0.765** |
| $2 \mapsto 4$ | 0.795 | 0.790 | 0.797 | **0.801** | 0.787 | 0.795 |
| $3 \mapsto 1$ | 0.780 | 0.774 | 0.761 | 0.770 | 0.733 | **0.812** |
| $3 \mapsto 2$ | 0.595 | 0.601 | 0.602 | 0.609 | 0.625 | **0.657** |
| $3 \mapsto 4$ | 0.817 | 0.819 | 0.815 | 0.818 | 0.824 | **0.836** |
| $4 \mapsto 1$ | 0.730 | 0.633 | 0.717 | 0.672 | 0.712 | **0.731** |
| $4 \mapsto 2$ | **0.640** | 0.591 | 0.635 | 0.583 | 0.637 | 0.635 |
| $4 \mapsto 3$ | 0.709 | 0.695 | 0.714 | 0.709 | 0.727 | **0.740** |
| Avg | 0.696 | 0.686 | 0.702 | 0.696 | 0.704 | **0.727** |

Higher is better. Best value in bold.

Table 17: Mortality prediction between various age groups from MIMIC to AUMC. Shown: mean AUROC over 10 random initializations.

| Sour $\mapsto$ Tar | w/o UDA | w/o CL and w/o NNCL | w/o CL | w/o NNCL | w/o Discriminator | CLUDA (ours) |
|---|---|---|---|---|---|---|
| $1 \mapsto 1$ | 0.736 | 0.710 | 0.734 | 0.731 | 0.757 | **0.782** |
| $1 \mapsto 2$ | 0.628 | 0.686 | 0.717 | 0.703 | 0.714 | **0.731** |
| $1 \mapsto 3$ | 0.662 | 0.670 | 0.677 | 0.685 | 0.692 | **0.707** |
| $1 \mapsto 4$ | 0.754 | 0.734 | 0.747 | 0.735 | **0.758** | 0.754 |
| $2 \mapsto 1$ | **0.835** | 0.823 | 0.803 | 0.829 | 0.803 | 0.822 |
| $2 \mapsto 2$ | 0.629 | 0.615 | 0.637 | 0.638 | 0.668 | **0.705** |
| $2 \mapsto 3$ | 0.656 | 0.645 | 0.691 | 0.679 | 0.709 | **0.714** |
| $2 \mapsto 4$ | 0.773 | 0.772 | 0.785 | 0.796 | 0.794 | **0.807** |
| $3 \mapsto 1$ | 0.763 | 0.778 | 0.777 | 0.775 | 0.771 | **0.789** |
| $3 \mapsto 2$ | 0.627 | 0.684 | 0.685 | 0.676 | 0.665 | **0.691** |
| $3 \mapsto 3$ | 0.711 | 0.723 | 0.744 | 0.731 | 0.745 | **0.751** |
| $3 \mapsto 4$ | 0.782 | 0.789 | 0.788 | **0.804** | 0.797 | 0.796 |
| $4 \mapsto 1$ | **0.714** | 0.641 | 0.635 | 0.708 | 0.648 | 0.689 |
| $4 \mapsto 2$ | 0.668 | 0.578 | **0.685** | 0.590 | 0.660 | 0.673 |
| $4 \mapsto 3$ | 0.619 | 0.577 | 0.602 | 0.589 | 0.604 | **0.635** |
| $4 \mapsto 4$ | 0.758 | 0.707 | 0.753 | 0.735 | 0.760 | **0.768** |
| Avg | 0.707 | 0.696 | 0.716 | 0.713 | 0.722 | **0.738** |

Higher is better. Best value in bold.

Table 18: Mortality prediction between various age groups from AUMC to MIMIC. Shown: mean AUROC over 10 random initializations.

| Sour $\mapsto$ Tar | w/o UDA | w/o CL and w/o NNCL | w/o CL | w/o NNCL | w/o Discriminator | CLUDA (ours) |
|---|---|---|---|---|---|---|
| $1 \mapsto 1$ | 0.693 | 0.718 | 0.718 | 0.728 | 0.744 | **0.791** |
| $1 \mapsto 2$ | 0.665 | 0.707 | 0.723 | 0.731 | 0.732 | **0.776** |
| $1 \mapsto 3$ | 0.609 | 0.618 | 0.625 | 0.630 | 0.612 | **0.679** |
| $1 \mapsto 4$ | **0.600** | 0.540 | 0.557 | 0.563 | 0.568 | 0.598 |
| $2 \mapsto 1$ | 0.703 | 0.722 | 0.745 | 0.747 | 0.739 | **0.780** |
| $2 \mapsto 2$ | 0.684 | 0.755 | 0.750 | 0.753 | 0.753 | **0.771** |
| $2 \mapsto 3$ | 0.641 | 0.681 | 0.682 | 0.682 | 0.683 | **0.702** |
| $2 \mapsto 4$ | 0.592 | 0.556 | 0.569 | 0.580 | 0.587 | **0.608** |
| $3 \mapsto 1$ | **0.805** | 0.762 | 0.764 | 0.785 | 0.761 | 0.785 |
| $3 \mapsto 2$ | 0.751 | 0.736 | 0.752 | 0.757 | 0.763 | **0.774** |
| $3 \mapsto 3$ | 0.720 | 0.716 | 0.719 | 0.713 | 0.723 | **0.729** |
| $3 \mapsto 4$ | **0.622** | 0.594 | 0.606 | 0.601 | 0.621 | 0.615 |
| $4 \mapsto 1$ | 0.801 | 0.793 | 0.804 | 0.806 | 0.808 | **0.819** |
| $4 \mapsto 2$ | 0.750 | 0.727 | 0.734 | 0.737 | 0.742 | **0.752** |
| $4 \mapsto 3$ | 0.709 | 0.664 | 0.684 | 0.687 | 0.691 | **0.711** |
| $4 \mapsto 4$ | 0.697 | 0.626 | 0.652 | 0.657 | 0.666 | 0.679 |
| Avg | 0.690 | 0.682 | 0.693 | 0.697 | 0.700 | **0.723** |

Higher is better. Best value in bold.

# I PREDICTION RESULTS OF MEDICAL PRACTICE

The main paper reported the average UDA performance between MIMIC and AUMC without the standard deviation of the results. Here, we provide the full results with gap filled (%) calculated for each method and additional AUPRC metric for decompensation and mortality predictions. Table 19 and Table 20 show the decompensation prediction results. Table 21 and Table 22 show the mortality prediction results. Table 23 show the length of stay prediction results.

Table 19: Decompensation prediction. Shown: AUROC (*mean ± std*) over 10 random initializations.

| Source | MIMIC | | AUMC | | Gap Filled (%) | |
| --- | --- | --- | --- | --- | --- | --- |
| Target | MIMIC | AUMC | AUMC | MIMIC | MIMIC | AUMC |
| w/o UDA | 0.831 ± 0.001 | 0.771 ± 0.004 | 0.813 ± 0.005 | 0.745 ± 0.004 | 0.0 | 0.0 |
| VRADAPurushotham et al. (2017) | 0.817 ± 0.002 | 0.773 ± 0.003 | 0.798 ± 0.003 | 0.764 ± 0.002 | +22.1 | +4.7 |
| CoDATSWilson et al. (2020) | 0.825 ± 0.003 | 0.772 ± 0.004 | 0.818 ± 0.005 | 0.762 ± 0.002 | +19.8 | +2.4 |
| AdvSKMLiu & Xue (2021) | 0.824 ± 0.002 | 0.775 ± 0.003 | 0.817 ± 0.004 | 0.766 ± 0.001 | +24.4 | +9.5 |
| CANKang et al. (2019) | 0.825 ± 0.002 | 0.773 ± 0.001 | 0.807 ± 0.004 | 0.740 ± 0.002 | −5.8 | +4.8 |
| CDANLong et al. (2018) | 0.824 ± 0.001 | 0.768 ± 0.003 | 0.817 ± 0.005 | 0.763 ± 0.005 | +20.9 | −7.1 |
| DDCTzeng et al. (2014) | 0.825 ± 0.001 | 0.772 ± 0.004 | 0.819 ± 0.004 | 0.765 ± 0.002 | +23.3 | +2.4 |
| DeepCORALSun & Saenko (2016) | 0.832 ± 0.002 | 0.774 ± 0.003 | 0.819 ± 0.004 | 0.768 ± 0.002 | +26.7 | +7.1 |
| DSANZhu et al. (2020) | 0.831 ± 0.002 | 0.774 ± 0.004 | 0.808 ± 0.004 | 0.759 ± 0.006 | +16.3 | +7.1 |
| HoMMChen et al. (2020a) | 0.829 ± 0.001 | 0.778 ± 0.004 | 0.816 ± 0.005 | 0.766 ± 0.001 | +24.4 | +16.7 |
| MMDARahman et al. (2020) | 0.821 ± 0.001 | 0.766 ± 0.003 | 0.814 ± 0.004 | 0.725 ± 0.006 | −23.3 | −11.9 |
| CLUDA (ours) | 0.832 ± 0.002 | **0.791 ± 0.004** | **0.825 ± 0.001** | **0.774 ± 0.002** | **+33.7** | **+47.6** |

Higher is better. Best value in bold. Black font: main results for UDA. Gray font: source ↦ source.

Table 20: Decompensation prediction. Shown: AUPRC (*mean ± std*) over 10 random initializations.

| Source | MIMIC | | AUMC | | Gap Filled (%) | |
| --- | --- | --- | --- | --- | --- | --- |
| Target | MIMIC | AUMC | AUMC | MIMIC | MIMIC | AUMC |
| w/o UDA | 0.240 ± 0.003 | 0.208 ± 0.003 | 0.214 ± 0.005 | 0.198 ± 0.004 | 0.0 | 0.0 |
| VRADAPurushotham et al. (2017) | 0.226 ± 0.003 | 0.209 ± 0.003 | 0.207 ± 0.002 | 0.184 ± 0.003 | −33.3 | +16.7 |
| CoDATSWilson et al. (2020) | 0.242 ± 0.003 | 0.213 ± 0.002 | 0.227 ± 0.002 | 0.211 ± 0.002 | +31.0 | +83.3 |
| AdvSKMLiu & Xue (2021) | 0.243 ± 0.002 | 0.215 ± 0.001 | 0.230 ± 0.005 | 0.214 ± 0.002 | +38.1 | +116.7 |
| CANKang et al. (2019) | 0.243 ± 0.001 | 0.215 ± 0.002 | 0.213 ± 0.004 | 0.166 ± 0.004 | −76.2 | +116.7 |
| CDANLong et al. (2018) | 0.240 ± 0.002 | 0.209 ± 0.002 | 0.231 ± 0.002 | **0.217 ± 0.003** | +45.2 | +16.7 |
| DDCTzeng et al. (2014) | 0.242 ± 0.001 | 0.214 ± 0.001 | 0.230 ± 0.001 | 0.211 ± 0.004 | +31.0 | +100.0 |
| DeepCORALSun & Saenko (2016) | 0.241 ± 0.002 | 0.216 ± 0.002 | 0.233 ± 0.002 | 0.213 ± 0.003 | +35.7 | +133.3 |
| DSANZhu et al. (2020) | 0.249 ± 0.002 | 0.216 ± 0.003 | 0.226 ± 0.002 | 0.174 ± 0.002 | −57.1 | +133.3 |
| HoMMChen et al. (2020a) | 0.241 ± 0.002 | 0.215 ± 0.003 | 0.230 ± 0.002 | 0.211 ± 0.001 | +31.0 | +116.7 |
| MMDARahman et al. (2020) | 0.241 ± 0.001 | 0.207 ± 0.004 | 0.227 ± 0.002 | 0.189 ± 0.002 | −21.4 | −16.7 |
| CLUDA (ours) | 0.253 ± 0.003 | **0.223 ± 0.003** | **0.239 ± 0.001** | 0.215 ± 0.002 | +40.5 | **+250.0** |

Higher is better. Best value in bold. Black font: main results for UDA. Gray font: source ↦ source.

Table 21: Mortality prediction. Shown: AUROC (*mean ± std*) over 10 random initializations.

| Source | MIMIC | | AUMC | | Gap Filled (%) | |
| --- | --- | --- | --- | --- | --- | --- |
| Target | MIMIC | AUMC | AUMC | MIMIC | MIMIC | AUMC |
| w/o UDA | 0.831 ± 0.001 | 0.709 ± 0.002 | 0.721 ± 0.005 | 0.774 ± 0.006 | 0.0 | 0.0 |
| VRADAPurushotham et al. (2017) | 0.827 ± 0.001 | 0.726 ± 0.005 | 0.729 ± 0.006 | 0.778 ± 0.002 | +7.0 | +141.7 |
| CoDATSWilson et al. (2020) | 0.832 ± 0.001 | 0.708 ± 0.005 | 0.724 ± 0.004 | 0.778 ± 0.004 | +7.0 | −8.3 |
| AdvSKMLiu & Xue (2021) | 0.830 ± 0.001 | 0.707 ± 0.001 | 0.724 ± 0.005 | 0.772 ± 0.004 | −3.5 | −16.7 |
| CANKang et al. (2019) | 0.830 ± 0.001 | 0.719 ± 0.002 | 0.715 ± 0.005 | 0.757 ± 0.004 | −29.8 | +83.3 |
| CDANLong et al. (2018) | 0.776 ± 0.001 | 0.716 ± 0.006 | 0.712 ± 0.004 | 0.772 ± 0.003 | −3.5 | +58.3 |
| DDCTzeng et al. (2014) | 0.831 ± 0.001 | 0.715 ± 0.005 | 0.721 ± 0.005 | 0.776 ± 0.003 | +3.5 | +50.0 |
| DeepCORALSun & Saenko (2016) | 0.832 ± 0.001 | 0.715 ± 0.005 | 0.727 ± 0.004 | 0.777 ± 0.003 | +5.3 | +50.0 |
| DSANZhu et al. (2020) | 0.832 ± 0.001 | 0.719 ± 0.006 | 0.721 ± 0.006 | 0.747 ± 0.007 | −47.4 | +83.3 |
| HoMMChen et al. (2020a) | 0.833 ± 0.001 | 0.707 ± 0.006 | 0.720 ± 0.005 | 0.778 ± 0.002 | +7.0 | −16.7 |
| MMDARahman et al. (2020) | 0.831 ± 0.001 | 0.718 ± 0.004 | 0.724 ± 0.006 | 0.773 ± 0.003 | −1.8 | +75.0 |
| CLUDA (ours) | 0.836 ± 0.001 | **0.739 ± 0.004** | **0.750 ± 0.001** | **0.789 ± 0.002** | **+26.3** | **+250.0** |

Higher is better. Best value in bold. Black font: main results for UDA. Gray font: source ↦ source.

Table 22: Mortality prediction. Shown: AUPRC (*mean ± std*) over 10 random initializations.

| Source | MIMIC | | AUMC | | Gap Filled (%) | |
|---|---|---|---|---|---|---|
| Target | MIMIC | AUMC | AUMC | MIMIC | MIMIC | AUMC |
| w/o UDA | 0.513 ± 0.004 | 0.412 ± 0.003 | 0.430 ± 0.002 | 0.427 ± 0.006 | 0.0 | 0.0 |
| VRADAPurushotham et al. (2017) | 0.501 ± 0.003 | 0.419 ± 0.003 | 0.422 ± 0.005 | 0.423 ± 0.006 | −4.7 | +38.9 |
| CoDATSWilson et al. (2020) | 0.518 ± 0.004 | 0.415 ± 0.002 | 0.435 ± 0.004 | 0.441 ± 0.004 | +16.3 | +16.7 |
| AdvSKMLiu & Xue (2021) | 0.518 ± 0.001 | 0.421 ± 0.001 | 0.441 ± 0.004 | 0.443 ± 0.004 | +18.6 | +50.0 |
| CANKang et al. (2019) | 0.830 ± 0.001 | 0.421 ± 0.002 | 0.436 ± 0.003 | 0.394 ± 0.006 | −38.4 | +50.0 |
| CDANLong et al. (2018) | 0.513 ± 0.001 | 0.422 ± 0.002 | 0.430 ± 0.001 | 0.435 ± 0.004 | +9.3 | +55.6 |
| DDCTzeng et al. (2014) | 0.520 ± 0.001 | 0.423 ± 0.003 | 0.432 ± 0.004 | 0.442 ± 0.004 | +17.4 | +61.1 |
| DeepCORALSun & Saenko (2016) | 0.520 ± 0.003 | 0.419 ± 0.002 | 0.432 ± 0.003 | 0.435 ± 0.005 | +9.3 | +38.9 |
| DSANZhu et al. (2020) | 0.514 ± 0.002 | 0.418 ± 0.005 | 0.435 ± 0.004 | 0.416 ± 0.007 | −12.8 | +33.3 |
| HoMMChen et al. (2020a) | 0.520 ± 0.002 | 0.418 ± 0.004 | 0.436 ± 0.004 | 0.448 ± 0.005 | +24.4 | +33.3 |
| MMDARahman et al. (2020) | 0.519 ± 0.002 | 0.419 ± 0.004 | 0.441 ± 0.003 | 0.440 ± 0.004 | +15.1 | +38.9 |
| CLUDA (ours) | **0.522 ± 0.002** | **0.428 ± 0.002** | 0.446 ± 0.003 | **0.452 ± 0.003** | **+29.1** | **+88.9** |

Higher is better. Best value in bold. Black font: main results for UDA. Gray font: source ↦ source.

Table 23: Length of stay prediction. Shown: KAPPA (*mean ± std*) over 10 random initializations.

| Source | MIMIC | | AUMC | | Gap Filled (%) | |
|---|---|---|---|---|---|---|
| Target | MIMIC | AUMC | AUMC | MIMIC | MIMIC | AUMC |
| w/o UDA | 0.178 ± 0.002 | 0.169 ± 0.003 | 0.246 ± 0.001 | 0.122 ± 0.001 | 0.0 | 0.0 |
| VRADAPurushotham et al. (2017) | 0.168 ± 0.003 | 0.161 ± 0.007 | 0.241 ± 0.002 | 0.126 ± 0.004 | +7.1 | −10.4 |
| CoDATSWilson et al. (2020) | 0.174 ± 0.002 | 0.159 ± 0.002 | 0.243 ± 0.001 | 0.120 ± 0.003 | −3.6 | −13.0 |
| AdvSKMLiu & Xue (2021) | 0.179 ± 0.002 | 0.172 ± 0.005 | 0.244 ± 0.002 | 0.123 ± 0.004 | +1.8 | +3.9 |
| CANKang et al. (2019) | 0.142 ± 0.003 | 0.173 ± 0.004 | 0.233 ± 0.001 | 0.118 ± 0.002 | −7.1 | +5.2 |
| CDANLong et al. (2018) | 0.176 ± 0.002 | 0.138 ± 0.004 | 0.244 ± 0.002 | 0.124 ± 0.002 | +3.6 | −40.3 |
| DDCTzeng et al. (2014) | 0.175 ± 0.001 | 0.163 ± 0.004 | 0.244 ± 0.001 | 0.123 ± 0.003 | +1.8 | −7.8 |
| DeepCORALSun & Saenko (2016) | 0.175 ± 0.002 | 0.166 ± 0.002 | 0.244 ± 0.001 | 0.126 ± 0.003 | +7.1 | −3.9 |
| DSANZhu et al. (2020) | 0.175 ± 0.002 | 0.154 ± 0.002 | 0.246 ± 0.001 | 0.122 ± 0.003 | 0.0 | −19.5 |
| HoMMChen et al. (2020a) | 0.174 ± 0.002 | 0.162 ± 0.006 | 0.243 ± 0.001 | 0.124 ± 0.001 | +3.6 | −9.1 |
| MMDARahman et al. (2020) | 0.158 ± 0.002 | 0.093 ± 0.004 | 0.246 ± 0.002 | 0.096 ± 0.004 | −46.4 | −98.7 |
| CLUDA (ours) | **0.216 ± 0.001** | **0.202 ± 0.006** | 0.276 ± 0.002 | **0.129 ± 0.003** | **+12.5** | **+42.9** |

Higher is better. Best value in bold. Black font: main results for UDA. Gray font: source ↦ source.

The results confirm our findings from the main paper: overall, our CLUDA achieves the best performance in both source and target domains.

## J ABLATION STUDY FOR MEDICAL PRACTICE

Here, we additionally provide our ablation study for the case study presented in Sec. 6.3. Specifically, Table 24 (source: MIMIC) and Table 25 (source: AUMC) evaluate the decompensation prediction. Table 26 (source: MIMIC) and Table 27 (source: AUMC) evaluate the mortality prediction. Table 28 (source: MIMIC) and Table 29 (source: AUMC) evaluate the length of stay prediction.

Table 24: Ablation study for decompensation prediction. Shown: AUROC (*mean $\pm$ std*) over 10 random initializations.

| Source | MIMIC | |
|---|---|---|
| Target | MIMIC | AUMC |
| w/o UDA | $0.831 \pm 0.001$ | $0.771 \pm 0.004$ |
| w/o CL and w/o NNCL ($\lambda_{CL} = 0, \lambda_{NNCL} = 0$) | $0.825 \pm 0.003$ | $0.772 \pm 0.004$ |
| w/o CL ($\lambda_{CL} = 0$) | $0.833 \pm 0.002$ | $0.782 \pm 0.003$ |
| w/o NNCL ($\lambda_{NNCL} = 0$) | $0.833 \pm 0.001$ | $0.786 \pm 0.003$ |
| w/o Discriminator ($\lambda_{disc} = 0$) | $\mathbf{0.841 \pm 0.001}$ | $0.787 \pm 0.003$ |
| CLUDA (ours) | $0.832 \pm 0.002$ | $\mathbf{0.791 \pm 0.004}$ |

Higher is better. Best value in bold. Black font: main results for UDA. Gray font: source $\mapsto$ source.

Table 25: Ablation study for decompensation prediction. Shown: AUROC (*mean $\pm$ std*) over 10 random initializations.

| Source | AUMC | |
|---|---|---|
| Target | AUMC | MIIV |
| w/o UDA | $0.813 \pm 0.005$ | $0.745 \pm 0.004$ |
| w/o CL and w/o NNCL ($\lambda_{CL} = 0, \lambda_{NNCL} = 0$) | $0.818 \pm 0.005$ | $0.761 \pm 0.003$ |
| w/o CL ($\lambda_{CL} = 0$) | $0.822 \pm 0.004$ | $0.763 \pm 0.003$ |
| w/o NNCL ($\lambda_{NNCL} = 0$) | $0.827 \pm 0.001$ | $0.771 \pm 0.004$ |
| w/o Discriminator ($\lambda_{disc} = 0$) | $\mathbf{0.832 \pm 0.002}$ | $0.771 \pm 0.002$ |
| CLUDA (ours) | $0.825 \pm 0.001$ | $\mathbf{0.774 \pm 0.002}$ |

Higher is better. Best value in bold. Black font: main results for UDA. Gray font: source $\mapsto$ source.

Table 26: Ablation study for mortality prediction. Shown: AUROC (*mean $\pm$ std*) over 10 random initializations.

| Source | MIMIC | |
|---|---|---|
| Target | MIMIC | AUMC |
| w/o UDA | $0.831 \pm 0.001$ | $0.709 \pm 0.002$ |
| w/o CL and w/o NNCL ($\lambda_{CL} = 0, \lambda_{NNCL} = 0$) | $0.830 \pm 0.002$ | $0.709 \pm 0.004$ |
| w/o CL ($\lambda_{CL} = 0$) | $0.836 \pm 0.001$ | $0.730 \pm 0.003$ |
| w/o NNCL ($\lambda_{NNCL} = 0$) | $0.840 \pm 0.002$ | $0.721 \pm 0.004$ |
| w/o Discriminator ($\lambda_{disc} = 0$) | $\mathbf{0.842 \pm 0.002}$ | $\mathbf{0.747 \pm 0.004}$ |
| CLUDA (ours) | $0.836 \pm 0.001$ | $0.739 \pm 0.004$ |

Higher is better. Best value in bold. Black font: main results for UDA. Gray font: source $\mapsto$ source.

Overall, the ablation study with different variants of our CLUDA confirms the importance of each component in our framework. Specifically, our CLUDA improves the prediction performance over all of its variants in all tasks except one (mortality prediction from MIMIC to AUMC). For this task, it is important to note that the best performing variant is w/o Discriminator, which has all the novel components of our CLUDA framework.

Table 27: Ablation study for mortality prediction. Shown: AUROC (*mean ± std*) over 10 random initializations.

| Source | AUMC | |
|---|---|---|
| Target | AUMC | MIMIC |
| w/o UDA | 0.721 ± 0.005 | 0.774 ± 0.006 |
| w/o CL and w/o NNCL ($\lambda_{CL} = 0, \lambda_{NNCL} = 0$) | 0.724 ± 0.004 | 0.778 ± 0.004 |
| w/o CL ($\lambda_{CL} = 0$) | 0.743 ± 0.001 | 0.781 ± 0.003 |
| w/o NNCL ($\lambda_{NNCL} = 0$) | 0.746 ± 0.004 | 0.781 ± 0.003 |
| w/o Discriminator ($\lambda_{disc} = 0$) | 0.749 ± 0.002 | 0.784 ± 0.004 |
| CLUDA (ours) | **0.750 ± 0.001** | **0.789 ± 0.002** |

Higher is better. Best value in bold. Black font: main results for UDA. Gray font: source ↦ source.

Table 28: Ablation study for length of stay prediction. Shown: KAPPA (*mean ± std*) over 10 random initializations.

| Source | MIMIC | |
|---|---|---|
| Target | MIMIC | AUMC |
| w/o UDA | 0.178 ± 0.002 | 0.169 ± 0.003 |
| w/o CL and w/o NNCL ($\lambda_{CL} = 0, \lambda_{NNCL} = 0$) | 0.173 ± 0.002 | 0.160 ± 0.005 |
| w/o CL ($\lambda_{CL} = 0$) | 0.212 ± 0.001 | 0.194 ± 0.009 |
| w/o NNCL ($\lambda_{NNCL} = 0$) | 0.212 ± 0.002 | 0.155 ± 0.003 |
| w/o Discriminator ($\lambda_{disc} = 0$) | 0.214 ± 0.001 | 0.196 ± 0.002 |
| CLUDA (ours) | **0.216 ± 0.001** | **0.202 ± 0.006** |

Higher is better. Best value in bold. Black font: main results for UDA. Gray font: source ↦ source.

Table 29: Ablation study for length of stay prediction. Shown: KAPPA (*mean ± std*) over 10 random initializations.

| Source | AUMC | |
|---|---|---|
| Target | AUMC | MIMIC |
| w/o UDA | 0.246 ± 0.001 | 0.122 ± 0.001 |
| w/o CL and w/o NNCL ($\lambda_{CL} = 0, \lambda_{NNCL} = 0$) | 0.242 ± 0.002 | 0.120 ± 0.003 |
| w/o CL ($\lambda_{CL} = 0$) | 0.271 ± 0.002 | 0.122 ± 0.003 |
| w/o NNCL ($\lambda_{NNCL} = 0$) | 0.274 ± 0.001 | 0.113 ± 0.001 |
| w/o Discriminator ($\lambda_{disc} = 0$) | 0.274 ± 0.001 | 0.125 ± 0.004 |
| CLUDA (ours) | **0.276 ± 0.002** | **0.129 ± 0.003** |

Higher is better. Best value in bold. Black font: main results for UDA. Gray font: source ↦ source.

# K CLUDA WITH ADAPTING OTHER CL METHODS

## K.1 CLUDA WITH SIMCLR

In our CLUDA framework, we capture contextual representation in time series data by leveraging contrastive learning. Specifically, we adapt momentum contrast (MoCo) (He et al., 2020) for contrastive learning in our framework. This choice is motivated by earlier research from other domains (He et al., 2020; Chen et al., 2020c; Yèche et al., 2021; Dwibedi et al., 2021), where MoCo was found to yield more stable negative samples (due to the momentum-updated feature extractor) as compared to other approaches throughout each training step, such as SimCLR(Chen et al., 2020b). In principle, stability yields stronger negative samples for the contrastive learning objectives and, therefore, increases the mutual information between the positive pair (i. e., two augmented views of the same sample). Furthermore, MoCo allows storing the negative samples within a queue, facilitating a larger number of negative samples for the contrastive loss as compared to SimCLR. As shown earlier (Bachman et al., 2019; Tian et al., 2020a;b), the lower bound of the mutual information between the positive pair increases with a larger number of negative samples in CL. With that motivation, we opted for MoCo (He et al., 2020) in our CLUDA instead of SimCLR(Chen et al., 2020b). Nevertheless, we evaluate our choice through numerical experiments below.

We now further perform an ablation study where we repeat the experiments with SimCLR (instead of MoCo) for our case study from Sec. 6.3. Specifically, we provide results for decompensation prediction (see Table 30), mortality prediction (see Table 31), and length of stay prediction (see Table 32).

Table 30: Decompensation prediction. Shown: AUROC (*mean $\pm$ std*) over 10 random initializations.

| Source | MIMIC | | AUMC | |
|---|---|---|---|---|
| Target | MIMIC | AUMC | AUMC | MIMIC |
| w/o UDA | 0.831 $\pm$ 0.001 | 0.771 $\pm$ 0.004 | 0.813 $\pm$ 0.005 | 0.745 $\pm$ 0.004 |
| CLUDA w/ SimCLR | 0.826 $\pm$ 0.001 | 0.776 $\pm$ 0.001 | 0.801 $\pm$ 0.005 | 0.751 $\pm$ 0.005 |
| CLUDA (ours) | 0.832 $\pm$ 0.002 | **0.791 $\pm$ 0.004** | 0.825 $\pm$ 0.001 | **0.774 $\pm$ 0.002** |

Higher is better. Best value in bold. Black font: main results for UDA. Gray font: source $\mapsto$ source.

Table 31: Mortality prediction. Shown: AUROC (*mean $\pm$ std*) over 10 random initializations.

| Source | MIMIC | | AUMC | |
|---|---|---|---|---|
| Target | MIMIC | AUMC | AUMC | MIMIC |
| w/o UDA | 0.831 $\pm$ 0.001 | 0.709 $\pm$ 0.002 | 0.721 $\pm$ 0.005 | 0.774 $\pm$ 0.006 |
| CLUDA w/ SimCLR | 0.827 $\pm$ 0.001 | 0.724 $\pm$ 0.004 | 0.748 $\pm$ 0.002 | 0.781 $\pm$ 0.002 |
| CLUDA (ours) | **0.836 $\pm$ 0.001** | **0.739 $\pm$ 0.004** | 0.750 $\pm$ 0.001 | **0.789 $\pm$ 0.002** |

Higher is better. Best value in bold. Black font: main results for UDA. Gray font: source $\mapsto$ source.

Table 32: Length of stay prediction. Shown: KAPPA (*mean $\pm$ std*) over 10 random initializations.

| Source | MIMIC | | AUMC | |
|---|---|---|---|---|
| Target | MIMIC | AUMC | AUMC | MIMIC |
| w/o UDA | 0.178 $\pm$ 0.002 | 0.169 $\pm$ 0.003 | 0.246 $\pm$ 0.001 | 0.122 $\pm$ 0.001 |
| CLUDA w/ SimCLR | 0.203 $\pm$ 0.001 | 0.178 $\pm$ 0.006 | 0.258 $\pm$ 0.005 | 0.107 $\pm$ 0.003 |
| CLUDA (ours) | 0.216 $\pm$ 0.001 | **0.202 $\pm$ 0.006** | 0.276 $\pm$ 0.002 | **0.129 $\pm$ 0.003** |

Higher is better. Best value in bold. Black font: main results for UDA. Gray font: source $\mapsto$ source.

The results confirm our choice for MoCo instead of SimCLR in capturing the contextual representation in time series. Specifically, our CLUDA improves the result of CLUDA w/ SimCLR in all tasks by a large margin. Despite being inferior to our CLUDA, CLUDA w/ SimCLR achieves better UDA performance compared to other baseline methods in decompensation prediction from MIMIC to AUMC, mortality prediction from AUMC to MIMIC, and length of stay prediction from MIMIC to AUMC. This shows the importance of leveraging the contextual representation into unsupervised domain adaptation. Besides, it highlights that our CLUDA can be further improved in the future with the recent advances in capturing the contextual representation of time series.

## K.2 CLUDA with NCL

We further compare our framework against neighborhood contrastive learning (NCL) (Yèche et al., 2021). For this, we replace the CL component (Sec. 4.3) of our CLUDA framework by another CL method called neighborhood contrastive learning (NCL) (Yèche et al., 2021). NCL also leverages the MoCo as in our CLUDA. It considers different time segments of the same subject as positive pairs (within a certain time window) when constructing the CL objective. NCL is specifically designed for the transfer learning setting, where the model is pre-trained on the unlabeled source domain and later fine-tuned on the smaller amount of labeled target domain. When labels are absent during the pre-training stage, NCL has been shown to be captured relevant signals in the embedding space for the downstream classification task.

However, our UDA setting is **different** from the transfer learning setting in Yèche et al. (2021): (a) UDA assumes the existence of source domain labels whereas transfer learning does not. (b) Transfer learning later leverages the labels of target domain whereas UDA does not require those labels. Since NCL's positive pairs may come from different classes (e.g., in healthcare, different time windows of a patient corresponding to different decompensation label or, in sensor datasets, different time windows of a subject corresponding to different activities from walking to running), we conjecture that it adds additional noise to the classifier network, leading to an inferior prediction performance.

Below we perform an ablation study where we replaced the CL component of our CLUDA with the CL of NCL. We kept all the other components the same (, i. e., discriminator, classifier networks, and our NNCL component). To select hyperparameters for NCL, we performed a grid-search analogous to the original implementation in Yèche et al. (2021). We provide the results for decompensation prediction (see Table 33), mortality prediction (see Table 34), and length of stay prediction (see Table 35).

The results confirm our conjecture that leveraging NCL in UDA setting leads to an inferior prediction performance. Specifically, our CLUDA performs significantly better than CLUDA w/NCL for all tasks and for both source and target domains. Notably, CLUDA w/NCL performs even worse than w/o UDA. For this, our explanation is that, since we have the labels of the source domain during the training time, the objectives of NCL and the classifier networks counteract each other. Therefore, our ablation study shows the need of tailoring the right contrastive learning objective for different problem settings (such as UDA vs. transfer learning). In sum, this confirms the effectiveness of our proposed framework architecture.

Table 33: Decompensation prediction. Shown: AUROC (*mean $\pm$ std*) over 10 random initializations.

| Source | MIMIC | | AUMC | |
|---|---|---|---|---|
| Target | MIMIC | AUMC | AUMC | MIMIC |
| w/o UDA | $0.831 \pm 0.001$ | $0.771 \pm 0.004$ | $0.813 \pm 0.005$ | $0.745 \pm 0.004$ |
| CLUDA w/ NCL | $0.774 \pm 0.003$ | $0.725 \pm 0.002$ | $0.763 \pm 0.003$ | $0.712 \pm 0.002$ |
| CLUDA (ours) | $0.832 \pm 0.002$ | $\mathbf{0.791 \pm 0.004}$ | $0.825 \pm 0.001$ | $\mathbf{0.774 \pm 0.002}$ |

Higher is better. Best value in bold. Gray font: source $\mapsto$ source. Other font: main results for UDA.

Table 34: Mortality prediction. Shown: AUROC (*mean ± std*) over 10 random initializations.

| Source | MIMIC | | AUMC | |
|---|---|---|---|---|
| Target | MIMIC | AUMC | AUMC | MIMIC |
| w/o UDA | 0.831 ± 0.001 | 0.709 ± 0.002 | 0.721 ± 0.005 | 0.774 ± 0.006 |
| CLUDA w/ NCL | 0.732 ± 0.003 | 0.677 ± 0.001 | 0.674 ± 0.002 | 0.705 ± 0.002 |
| CLUDA (ours) | **0.836 ± 0.001** | **0.739 ± 0.004** | **0.750 ± 0.001** | **0.789 ± 0.002** |

Higher is better. Best value in bold. Gray font: source ↦ source. Other font: main results for UDA.

Table 35: Length of stay prediction. Shown: KAPPA (*mean ± std*) over 10 random initializations.

| Source | MIMIC | | AUMC | |
|---|---|---|---|---|
| Target | MIMIC | AUMC | AUMC | MIMIC |
| w/o UDA | 0.178 ± 0.002 | 0.169 ± 0.003 | 0.246 ± 0.001 | 0.122 ± 0.001 |
| CLUDA w/ NCL | 0.141 ± 0.002 | 0.132 ± 0.001 | 0.173 ± 0.003 | 0.080 ± 0.002 |
| CLUDA (ours) | **0.216 ± 0.001** | **0.202 ± 0.006** | **0.276 ± 0.002** | **0.129 ± 0.003** |

Higher is better. Best value in bold. Gray font: source ↦ source. Other font: main results for UDA.

## L  DISCUSSION FOR VARIABLE-LENGTH TIME SERIES

Following earlier works of UDA for time series (Cai et al., 2021; Liu & Xue, 2021; Wilson et al., 2020; 2021), we defined the problem (see Sec. 3) in way that each time series has a fixed length $T$. In case the length of time series differs too much within a dataset or when the entire history of time series needs to be considered, it may be preferred to account for variable-length time series. Here, we briefly discuss how our CLUDA can be adapted to variable-length time series inputs. We further believe that our discussion below may also be applicable for existing UDA baselines (with minor modifications). One can adapt our CLUDA primarily in two different ways.

**(a) Straightforward approach:** One can configure a temporal convolutional network (TCN) (Bai et al., 2018) as feature extractor to handle the longest time-series and pre-pad the shorter ones with a certain value. Then, the output of the feature extractor is used analogous to our original CLUDA framework. However, in case of too long and too short time series being present in the same dataset, we suspect TCN may not capture meaningful representations for the short time series due to pre-padded values (e. g., zeros) being dominant. If the length of time series varies by the order of dilation factor (e. g., a number of 1x, 2x, 4x, 8x time steps with dilation factor 2 of TCN), one can extract the features from the earlier layers of TCN for the shorter time series (thereby basically considering the receptive field). This way, one could avoid the dominance of pre-padded values.

**(b) Tailored approach:** Variable-length time series can be naturally modeled by a generative neural network such as variational recurrent neural network (VRNN) (Chung et al., 2015) or deep Markov model (DMM) (Krishnan et al., 2017). As such, one can leverage the latent variables of the generative model as input to our contrastive learning component of CLUDA. Here, we hypothesize that using individual latent variables (of each time step) as input to CL would be (i) computationally too expensive and (ii) not so meaningful, since we apply augmentations to the entire time series and not to individual time steps. Therefore, we suggest an attention module which will process the sequence of latent variables and output the aggregated latent representation of entire time series. The output of the attention module can then be used in our CLUDA framework by multiple components, such as CL, NNCL, and the classifier network. To summarize, our suggestion as a short recipe: One can (1) get the latent variables from a generative model, (2) aggregate them via an attention module, and (3) use the output as the output of the feature extractor as in our original CLUDA framework.

