# OpenReview forum: "Contrastive Learning for Unsupervised Domain Adaptation of Time Series"
_ICLR.cc/2023/Conference — ICLR 2023 poster_

### Official Review · Reviewer_ZBbo · 2022-10-24

**Confidence:** 4
**Correctness:** 3
**Technical Novelty And Significance:** 3
**Empirical Novelty And Significance:** 3
**Recommendation:** 8

**Clarity, Quality, Novelty And Reproducibility:**

The work is well written, and the contributions are clearly laid out, the results are interesting. The anonymous link to the code is made available.

**Strength And Weaknesses:**

### Strengths
+  Well motivated
+ Well written and paper is well organized.
+ Extensive applications
+ Ablation study

### Weaknesses
- While the paper tackles time series, it seems that the model can only handle same length time series in the problem formulation (see comments)
- Some recent works on UDA for time series (video analytics) which also use contrastive learning are not compared to/referred to (see comments)
- the motivation for design of certain components can benefit from further explanation (see comments)
- Some components of the overall model are not well explained (see comments)


### Comments/Questions:

1. Can the model be applied to different length time-series? If yes, how will the architecture change?
2. [1] (CoMix) Also uses a similar contrastive learning approach for temporal data in Video domain adaptation. How does the method compare to that work. Consider including it in your literature review.
3. While the authors use certain components like adversarial training, contrastive learning via certain loss functions, and gradient reversal layers. This combination and choice of each of these components is interesting. Can the authors justify why these were chosen, and what were the other alternatives? With some addition motivation on how these come together in the optimization?
4. Figure 2 is perhaps the most difficult to parse. For instance, it seems that $x_k^s$ is transformed to $z_k^s$ (bottom left corner) without any transfortmation function? Also it is unclear what "queue" is here. I recommed redrafting this figure, with additional labels for multiple components listed here, and then describing them in the caption and in the write-up.


[1] Sahoo (2021). Contrast and Mix: Temporal Contrastive Video Domain Adaptation with Background Mixing, NeurIPS 2021.

**Summary Of The Paper:**

The paper proposes a contrastive learning and adversarial training-based approach to accomplish domain adaptation for time series data. The paper is rooted in strong motivations in healthcare where such a transfer is important for reliable operations. The results are convincing and the ablations are much appreciated.

**Summary Of The Review:**

While I list these weaknesses, I think the authors can address these during the rebuttal period, and I look forward to hearing from them.

---

> ### Author Response · Authors · 2022-11-14
> **Response to Reviewer ZBbo (1/3)**
>
> We thank you so much for your feedback and your questions. As a result, we improved our paper in several ways:
>
> 1. We now discuss how our CLUDA framework can be applied to variable-length time series (see new Appendix L).
>
> 2. We expanded our related work and also covered UDA for videos. We further discussed how this is different from our work and thus how ours is novel (see revised Appendix A).
>
> 3. We further motivate our chosen architecture and discuss how the selected components facilitate better performance (see revised Result section and see new Discussion).
>
> 4. We made our overview of the CLUDA framework easier to understand. Specifically, we re-drew our figure (Figure 1) and re-wrote Section 4.1 introducing our framework.
>
> Overall, we believe our work has been greatly improved by following your comments. Below, you can find our detailed answers for all the points you raised.
>
> ## Responses to “Comments/Questions”
>
>
> **Can the model be applied to different length time-series? If yes, how will the architecture change?**
>
> We thank you for this interesting and important question. Indeed, one can leverage our CLUDA framework in two different ways:
>
> a) Straightforward approach: one can configure a temporal convolutional network (TCN) [1] as feature extractor to handle the longest time-series and pre-pad the shorter ones with a certain value. Then the output of the feature extractor is used as before in our framework. However, in case of too long and too short time series being present, we suspect that the TCN might not capture meaningful representations for the short time series, due to pre-padding values (e.g. zeros) being dominant.
> (Note: if the length of time series varies by the order of dilation factor, e.g. 1x, 2x, 4x, 8x time steps with dilation factor 2 of TCN, one can extract the features from the earlier layers of TCN for the shorter time series (basically considering the receptive field). This way, one could avoid the dominance of pre-padding values. However, we are not aware of any such implementation and therefore how it works in practice.)
>
> b) Tailored approach: variable-length time series data can be modeled by a generative neural network such as variational recurrent neural network (VRNN) [2] or deep Markov model (DMM) [3]. As such, one can leverage the latent variables of the generative model as input to our contrastive learning framework. We hypothesize that using individual latent variables (of each time step) would be (i) computationally too expensive and (ii) not so meaningful since we apply augmentations to entire time series, not to individual time steps. Therefore, we would suggest an attention module which will process the sequence of latent variables and output the aggregated latent representation of the entire time series. The output of the attention module can then be used in our CLUDA framework by multiple components, such as CL, NNCL, and classifier. To summarize our suggestion as a short recipe: One can (1) get the latent variables from a generative model, (2) aggregate them via an attention module, and (3) use this output as the output of the feature extractor as in our original CLUDA framework.
>
> We believe the discussion of your question is useful for the community. This is why we added a new discussion section (Appendix L) explaining how to apply CLUDA to variable-length time series. Similar strategies can be applied for other baselines as well.
>
>
>
> **Comix also uses a similar contrastive learning approach for temporal data in Video domain adaptation. How does the method compare to that work? Consider including it in your literature review.**
>
> We thank you for your valuable suggestion to include CoMix in the literature review. We followed your suggestion closely and updated our literature review accordingly. Specifically, we discuss several approaches for UDA in video and highlight the differences to ours (see our updated Appendix A).
>
> For your convenience, we would like to elaborate some of the key differences between CoMix and our CLUDA framework below (these are also included in our new Appendix A).
>
> (i) CoMix also leverages contrastive learning in their framework. However, for CL, they rely on different speeds of the same video to generate positive pairs which is not applicable in our setting because it changes the characteristics of time series. For instance, in our sensor datasets, changing the speed can make walking and running indistinguishable, or,  in our medical dataset, can make different disease progressions indistinguishable. Instead, for that purpose, we employ augmentation techniques that do not drastically change the frequency components of the original time series.

---

> > ### Author Response · Authors · 2022-11-14
> > **Response to Reviewer ZBbo (2/3)**
> >
> > (ii) For the domain alignment, CoMix proposes background mixing. In plain terms, they first create *synthetic* videos (by a convex combination of source and target frames) and bring source (target) representations closer to the synthetic video representations. In our work, our domain alignment component NNCL *directly* works on the original representations; hence, ours does not require any synthetic data generation. Of note, their approach of convex combination is not straightforward to apply to time series, since the superposition of two time series can lead to quite different characteristics. As another note, CoMix introduces their domain alignment strategy as a replacement to adversarial training. Since we found that adversarial training facilitates consistent performance gains (yet in smaller magnitude compared to other components), it would be interesting to see if adversarial training further improves the performance of CoMix.
> >
> > (iii) CoMix proposes a contrastive learning approach that requires pseudolabels of the target samples. It introduces additional complexity of finding appropriate confidence and the threshold for the estimated target labels, while the classifier network is being updated at each training step. In comparison, our CLUDA framework does not require any pseudolabeling. Of note, we implemented one baseline (CAN [4]) that requires pseudolabels of the target domain but led to an inferior performance. We attribute the inferior performance to the need of a tailored pseudolabeling strategy.
> >
> > (iv) CoMix has been trained in two stages. It is first pre-trained on the source domain and then trained by a custom loss function. In comparison, our CLUDA is directly trained end-to-end. Importantly, our CLUDA achieves state-of-the-art performance in our task.
> >
> > **While the authors use certain components like adversarial training, contrastive learning via certain loss functions, and gradient reversal layers. This combination and choice of each of these components is interesting. Can the authors justify why these were chosen, and what were the other alternatives? With some additional motivation on how these come together in the optimization?**
> >
> > Thank you for your interest in our model architecture and giving us the opportunity to discuss our chosen components.
> >
> > We have started our initial experiments with adversarial training and gradient reversal layer (which enables adversarial training) with no contrastive learning. In fact, this is the reason why we introduced our CLUDA framework starting with adversarial training (Section 4.2). We have observed that adversarial training facilitates matching source and target distributions (i.e. overlapping point clouds of two domains) but it doesn’t facilitate matching label-specific distributions across domains. This observation is supported by several examples such as Figure 10 in DANN [5], Figures 5 and 6 of VRADA [6], and also Figure 11 in our Appendix where we compare all methods, including the adversarial training. The essence is that source and target domains span the same space, but the classifier trained on the source domain does not necessarily perform well on the target domain. We now highlight this observation more extensively in our *insights* paragraph in Section 6.1.
> >
> > Motivated by the above reasoning, we hypothesized a two-phased strategy to improve upon earlier works:
> >
> > (1) Can we learn label-preserving representations for both domains from the feature extractor? This way, we enable adversarial training to align the representations *of each label/class* across domains. For this, we leverage the contrastive learning [7] framework on both domains, which is observed to capture label-preserving information from the context [8,9,10,11,12,13,14]. Thereby, the representations of the same labels get closer across domains. However, the decision boundary of the classifier can still miss some of the target domain samples, since the classifier is prone to overfit to the source domain in high dimensional representation space. This led us to the following question.
> >
> > (2) Can we further align the individual samples across domains? This way, each target sample is further brought closer to its most similar source domain-counterpart. We achieve this via a nearest-neighborhood CL [15]. Therefore, during the evaluation, the classifier sees target representations which are similar to the source representations from the training time. (We refer to Fig. 11 for the visual comparison of how the samples of the same label are pulled together). This leads to a better prediction accuracy on the target domain.
> >
> > Overall, we would like to note that we find our model components complementary to each other. This is supported by the ablation studies for all of our experiments.

---

> > > ### Author Response · Authors · 2022-11-14
> > > **Response to Reviewer ZBbo (3/3)**
> > >
> > > We believe our CLUDA can be further improved by the future advances in representation learning of time series, which could be used in place of our current CL component (as discussed in Appendix K.1). For our NNCL component, one can design novel neighborhood selection strategies to improve upon our NNCL.
> > >
> > > For the optimization wrt. all these loss components, our CLUDA framework is trained in an end-to-end manner, i.e. it doesn’t require any additional pre-training and/or fine-tuning steps. The weights of the different loss components can be tuned as hyperparameters. For the hyperparameter tuning, we suggest considering the scale of each loss component: for instance, contrastive learning loss tends to increase with the number of negative samples. In our CLUDA framework, our CL component (Section 4.3) has $J$ negative samples (size of queue), which is larger than the negative samples $N_s$ of our NNCL component (Section 4.4). Therefore, to counter-balance this effect, one can start with smaller weight for CL (compared to NNCL) as a simple heuristic.
> > >
> > > => To improve our paper, we expanded our Discussion in Section 7. Therein, we elaborate more extensively why our framework is superior and which components contribute to the performance gains.
> > >
> > > **Figure 2 is perhaps the most difficult to parse. For instance, it seems that
> > > x_k^s is transformed to z_k^s (bottom left corner) without any transformation function? Also it is unclear what "queue" is here. I recommend redrafting this figure, with additional labels for multiple components listed here, and then describing them in the caption and in the write-up.**
> > >
> > > Thank you for your suggestions to make our figure more informative. We followed your idea and revised our figure (previously Figure 2, now Figure 1 in the revised manuscript). Specifically, we used different coloring for the output of different stages, making them easier to identify. We added labels to each component. We used a dashed box to highlight each loss function and addressed which section of the paper they belong. We also re-wrote the caption of the figure and provided a brief description of our CLUDA framework. We clarified the queue and also added more details to our paper. We further now show the transformation $\tilde{F(\cdot)}$ for the source domain: $x_k^s$ is actually transformed to $z_k^s$ via a momentum updated feature extractor. Finally, we made our Section 4.1 consistent with the new figure.
> > >
> > > ## References
> > >
> > > [1] Shaojie Bai, J Zico Kolter, and Vladlen Koltun. An empirical evaluation of generic convolutional and recurrent networks for sequence modeling. arXiv preprint arXiv:1803.01271, 2018.
> > >
> > > [2] Junyoung Chung, Kyle Kastner, Laurent Dinh, Kratarth Goel, Aaron C Courville, and Yoshua Bengio. A recurrent latent variable model for sequential data. NeurIPS, 2015.
> > >
> > > [3] Rahul Krishnan, Uri Shalit, and David Sontag. Structured inference networks for nonlinear state space models. In AAAI, 2017.
> > >
> > > [4] Guoliang Kang, Lu Jiang, Yi Yang, and Alexander G Hauptmann. Contrastive adaptation network for unsuper-vised domain adaptation. In CVPR, 2019.
> > >
> > > [5] Yaroslav Ganin, Evgeniya Ustinova, Hana Ajakan, Pascal Germain, Hugo Larochelle, François Laviolette, Mario Marchand, and Victor Lempitsky. Domain-adversarial training of neural networks. JMLR,  2016.
> > >
> > > [6] Sanjay Purushotham, Wilka Carvalho, Tanachat Nilanon, and Yan Liu. Variational recurrent adversarial deep domain adaptation. In ICLR, 2017.
> > >
> > > [7] Kaiming He, Haoqi Fan, Yuxin Wu, Saining Xie, and Ross Girshick. Momentum contrast for unsupervised visual representation learning. In CVPR, 2020.
> > >
> > > [8] Philip Bachman, R Devon Hjelm, and William Buchwalter. Learning representations by maximizing mutual information across views. NeurIPS, 2019.
> > >
> > > [9] Ting Chen, Simon Kornblith, Mohammad Norouzi, and Geoffrey Hinton. A simple framework for contrastive learning of visual representations. In ICML, 2020.
> > >
> > > [10] Xinlei Chen, Haoqi Fan, Ross Girshick, and Kaiming He. Improved baselines with momentum contrastive learning. arXiv preprint arXiv:2003.04297,2020.
> > >
> > > [11] Songwei Ge, Shlok Mishra, Chun-Liang Li, Haohan Wang, and David Jacobs. Robust contrastive learning using negative samples with diminished semantics. NeurIPS, 2021.
> > >
> > > [12] Jean-Bastien Grill, Florian Strub, Florent Altché, Corentin Tallec, Pierre Richemond, Elena Buchatskaya, Carl Doersch, Bernardo Avila Pires, Zhaohan Guo, Mohammad Gheshlaghi Azar, et al. Bootstrap your own latent–a new approach to self-supervised learning. NeurIPS, 2020.
> > >
> > > [13] Yonglong Tian, Dilip Krishnan, and Phillip Isola. Contrastive multiview coding. In ECCV, 2020.
> > >
> > > [14] Yonglong Tian, Chen Sun, Ben Poole, Dilip Krishnan, Cordelia Schmid, and Phillip Isola. What makes for good views for contrastive learning? NeurIPS, 2020.
> > >
> > > [15] Debidatta Dwibedi, Yusuf Aytar, Jonathan Tompson, Pierre Sermanet, and Andrew Zisserman. With a little help from my friends: Nearest-neighbor contrastive learning of visual representations. In ICCV, 2021.

---

### Official Review · Reviewer_X1Eq · 2022-10-25

**Confidence:** 3
**Correctness:** 4
**Technical Novelty And Significance:** 3
**Empirical Novelty And Significance:** 3
**Recommendation:** 8

**Clarity, Quality, Novelty And Reproducibility:**

Clarity: Clear.
Quality: Good.
Novelty: As far as I can tell, novel.


**Strength And Weaknesses:**

Strenghts:
- The paper is clear, well-written and motivated.
- The empirical results are strong: the model appears to outperform baselines conclusively.

Weaknesses:
x No major weaknesses found.

Edit: to add on this comment, I have responded to the post titled "global response to reviewers".

**Summary Of The Paper:**

The authors propose a framework for unsupervised domain adaptation of time-series.Their approach has 3 components, a feature extractor, discriminator and classifier. Essentially the framework is a combination of the usual classifier/discriminator setup used in domain adaptation with an unsupervised learning objective based on MoCo.

**Summary Of The Review:**

I found multiple positive points in this submission, as listed above. The idea seems straightforward enough that I suspect it might have occured independently in other fields (such as computer vision) but was unable to find a reference. For this point I will defer to another reviewer more experienced in unsupervised domain adaptation.

---

> ### Author Response · Authors · 2022-11-14
> **Response to Reviewer X1Eq**
>
> We thank you very much for your positive evaluation! We would like to kindly point out that contrastive learning has been introduced in the literature, but it was not tailored to UDA for time series. This presents our main contribution. In particular, we show (a) how to leverage CL for capturing the contextual representation of time series within each domain and (b) how to further align these representations across domains.

---

### Official Review · Reviewer_xEBG · 2022-10-25

**Confidence:** 3
**Correctness:** 1
**Technical Novelty And Significance:** 2
**Empirical Novelty And Significance:** 2
**Recommendation:** 3

**Clarity, Quality, Novelty And Reproducibility:**

The paper is not well-written and it is hard to follow at some places. I have serious doubts about novelty and reproducibility.

**Strength And Weaknesses:**

Strengths:
1-	Domain adaptation for Time series data is not much directly addressed in the literature.

Weakness:
1-	The authors fail to motivate or justify the approach taken in this work
2-	Some important baselines are missing.
3-	The paper is difficult to follow sometimes because it is not written well.


**Summary Of The Paper:**

The authors aim to propose a framework for unsupervised domain adaptation of time series by doing a weird combination of existing models without good justification/motivation for what they introduce.  Specifically, they combine the model introduced in “domain adversarial training of neural networks” by a type of contrastive learning model.

**Summary Of The Review:**

Not sure if I understand the authors claim that the existing approaches for this problem from computer vision cannot be applied to time series. Time series and images are very similar, and the difference is that one is 1D and the other is 2D. That is why one can simply adapt any convolutional based model from images to time series. The authors need to better explain why the existing models for vision cannot be adapted for time series.

Another unsupported claim by the author: existing works for UDA of time series merely align the features across source and target domain.

Not sure how the approach introduced in this paper is different from the one in “domain adversarial training of neural networks” work? It seems exactly the same formulation and approach. Figure 1 in this paper is exactly the same as Figure 1 in that paper. Equation 1-2 in this paper are also components of Equation 18 in that paper. The idea seems to be exactly the same thing. The authors combined this later in Section 4.4. with a nearest neighbor contrastive learning. However, they do not motivate the need for this approach in the paper. The authors also fail to explain the details of Figure 2. I am very much confused by this extra loss function as the loss function in Equation 3 aims to make sure that the learned embedding is good for both domains.  Interestingly, the authors do not compare their results with “domain adversarial training of neural networks”, i.e. Figure 1, either.

Some examples of writing problems:
Beginning of Section 2: while pushing dissimilar samples  while pushing dissimilar samples away.
Beginning of Section 3 and 7: An abbreviation for Unsupervised domain adaptation is already introduced.

---

> ### Author Response · Authors · 2022-11-14
> **Response to Reviewer xEBG (1/2)**
>
> We thank you for your time and for providing constructive feedback. Following your comments, we improved our paper in several ways:
>
> 1. We have motivated the need for our framework and discussed why our CLUDA framework works better than the state-of-the-art. For this, we have revised our Introduction, Related Work, and Results sections. We further added a new Discussion section.
>
> 2. We compare our CLUDA against adversarial training, finding that CLUDA is superior. Further, we added new experiments to show the importance of the CL method (see new Appendix K).
>
> 3. We improved our presentation. In particular, we re-drew the figure of our architecture and re-wrote Section 4.1 introducing our architecture. We also improved the writing throughout our paper.
>
> We are confident that our work has greatly improved as a result. Below, we respond to your comments more in detail.
>
> ## Responses to “Weaknesses”
>
> **The authors fail to motivate or justify the approach taken in this work**
>
> Thank you for your feedback. We have made several improvements in our paper to better motivate and justify our approach. (1) We revised our wording in our Introduction and our Related Work sections. (2) We reworked Figure 1 where we visualized our architecture and now show each component explicitly. (3) We expanded our discussion explaining why our framework is superior and, therein, link to the contributions of each framework component.
>
> **Some important baselines are missing**
>
> Thank you for your comment regarding our experimental setup. Following your “Summary of the review”, we believe this comment is related to the comparison of our CLUDA against domain adversarial training.
>
> We report results for domain adversarial learning as follows: we use “w/o CL and w/o NNCL'' to refer to our benchmark based on domain adversarial training. Therein, the other components of CLUDA have now been disabled (introduced in Section 4.2). We added more materials to emphasize this comparison in our main paper: for example, in Sec 4.1, we now explicitly state that we have baselines comparing our CLUDA against this architecture. In addition, we introduce w/o CL and w/o NNCL in our ablation study (Section 6.1). Therein, we now iterate in greater depth that it corresponds to domain adversarial training.
>
> Further, following the comments from the other reviewers, we additionally added a new version of our CLUDA with NCL adaptation (see [1] for NCL). The results of this experiment can be found in Appendix K.2.
>
> **The paper is difficult to follow sometimes because it is not written well**
>
> Thank you for your comment regarding the flow of our paper. We have improved our writing in several ways. (1) We have carefully edited our paper for clarity and removed typos. (2) We reworked Figure 1 to visualize our framework. (3) We re-wrote Section 4.1 to explain our CLUDA framework and connect directly to our Figure 1.
>
> ## Responses to “Summary Of The Review”
>
> **The authors need to better explain why the existing models for vision cannot be adapted for time series.**
>
> Thank you. After reading your comment, we came to the realization that we need to revise our wording. Indeed, baselines from computer vision are — in principle — applicable but not tailored to time series data because of which they are inferior. We have carefully revised our wording as a result.
>
> **Another unsupported claim by the author: existing works for UDA of time series merely align the features across source and target domain.**
>
> Thanks. We welcome the opportunity for clarification. Earlier works extract the features of time series and directly align these extracted features. Some works achieve this via adversarial training [2,3], whereas others achieve such alignment via minimizing the discrepancy between two domains [4,5]. As a result, the domain distribution becomes indistinguishable (i.e. overlapping point clouds of two domains) but it doesn’t facilitate matching of label-specific distributions across domains. This can be observed in earlier works (Figure 10 in DANN [6], Figures 5 and 6 in VRADA [3]), as well as in Figure 11 in our Appendix. Accordingly, our main improvement over existing works is to align the representations *of each label/class* across domains. For this, we leverage the contrastive learning [7] framework on both domains, which is observed to capture label-preserving information from the context [8,9,10,11,12,13,14].
>
> We followed your advice closely and clarified the above in our paper (e.g., we rewrote parts of our Related Work section, the research gap, and our Discussion section). In particular, we supported our claims by adding a new discussion paragraph in Section 6.1.

---

> > ### Author Response · Authors · 2022-11-14
> > **Response to Reviewer xEBG (2/2)**
> >
> > **It seems exactly the same formulation and approach with  “domain adversarial training of neural networks”**
> >
> > Thank you for your comment. We would like to clarify the difference between domain adversarial training and our work. In our framework, domain adversarial training can be seen as a special variant of CLUDA where the loss components of CL and NNCL are completely discarded. For all of our experiments, we performed an ablation study to highlight the importance of the model components. We find that our framework is consistently superior.
> >
> > Further, we explicitly compared our CLUDA against domain adversarial training, which we represented as “w/o CL and w/o NNCL”. This comparison can be found in different sections: (i) performance on benchmark datasets (Section 6.1), (ii) the more detailed ablation study on the benchmark datasets (Appendix F), (iii) the ablation study across various age groups (Appendix H), and (iv) the ablation study across health institutions (Appendix J). Again, we find consistent support that our framework is superior by a large margin.
> >
> > To make the above clear for our readers, we added additional explanations in Section 4.1 when providing an overview of our CLUDA. Therein, we now wrote explicitly: domain adversarial training is different from our framework for the above reason. We further compare our framework against the adversarial training. We additionally iterate in Section 6.1 that w/o CL and w/o NNCL corresponds to domain adversarial training, finding a large performance gain due to our framework.
> >
> > **The authors also fail to explain the details of Figure 2.**
> >
> > Thank you for your feedback to make our Figure more detailed and more self-explanatory  (previously Figure 2, now Figure 1 in the revised manuscript). Following your comment we made the following improvements:
> >
> > We added colors to our components and embeddings. We made it easier to identify the same embeddings used by several components of our CLUDA framework.
> > We added labels to each component and also added the relevant section numbers to make it easier to locate in our paper.
> > We revised our Section 4.1 and directly linked it to our new figure.
> > We re-wrote the caption of the figure and provided a brief overview of our CLUDA framework with references to the relevant sections from our paper.
> >
> > The new figure can now be found as Fig. 1 in our updated submission. We hope the new version is more comprehensible and better aligned with our paper.
> >
> >
> > **Some examples of writing problems**
> >
> > We thank the reviewer for the suggestions. We have revised our writing accordingly.
> >
> > ## References
> >
> > [1] Hugo Yèche, Gideon Dresdner, Francesco Locatello, Matthias Hüser, and Gunnar Rätsch. Neighborhood contrastive learning applied to online patient monitoring. In ICML, 2021.
> >
> > [2] Garrett Wilson, Janardhan Rao Doppa, and Diane J Cook. Multi-source deep domain adaptation with weak supervision for time-series sensor data. In KDD, 2020.
> >
> > [3] Sanjay Purushotham, Wilka Carvalho, Tanachat Nilanon, and Yan Liu. Variational recurrent adversarial deep domain adaptation. In ICLR, 2017.
> >
> > [4] Qiao Liu and Hui Xue. Adversarial spectral kernel matching for unsupervised time series domain adaptation. In IJCAI, 2021.
> >
> > [5] Ruichu Cai, Jiawei Chen, Zijian Li, Wei Chen, Keli Zhang, Junjian Ye, Zhuozhang Li, Xiaoyan Yang, and Zhenjie Zhang. Time series domain adaptation via sparse associative structure alignment. In AAAI, 2021.
> >
> > [6] Yaroslav Ganin, Evgeniya Ustinova, Hana Ajakan, Pascal Germain, Hugo Larochelle, François Laviolette, Mario Marchand, and Victor Lempitsky. Domain-adversarial training of neural networks. JMLR, 17:2096–2030, 2016.
> >
> > [7] Kaiming He, Haoqi Fan, Yuxin Wu, Saining Xie, and Ross Girshick. Momentum contrast for unsupervised visual representation learning. In CVPR, 2020.
> >
> > [8] Philip Bachman, R Devon Hjelm, and William Buchwalter. Learning representations by maximizing mutual information across views. NeurIPS, 2019.
> >
> > [9] Ting Chen, Simon Kornblith, Mohammad Norouzi, and Geoffrey Hinton. A simple framework for contrastive learning of visual representations. In ICML, 2020.
> >
> > [10] Xinlei Chen, Haoqi Fan, Ross Girshick, and Kaiming He. Improved baselines with momentum contrastive learning. arXiv preprint arXiv:2003.04297, 2020.
> >
> > [11] Songwei Ge, Shlok Mishra, Chun-Liang Li, Haohan Wang, and David Jacobs. Robust contrastive learning using negative samples with diminished semantics. NeurIPS, 2021.
> >
> > [12] Jean-Bastien Grill, Florian Strub, Florent Altché, Corentin Tallec, Pierre Richemond, Elena Buchatskaya, Carl Doersch, Bernardo Avila Pires, Zhaohan Guo, Mohammad Gheshlaghi Azar, et al. Bootstrap your own latent – a new approach to self-supervised learning. NeurIPS, 2020.
> >
> > [13] Yonglong Tian, Dilip Krishnan, and Phillip Isola. Contrastive multiview coding. In ECCV, 2020.
> >
> > [14] Yonglong Tian, Chen Sun, Ben Poole, Dilip Krishnan, Cordelia Schmid, and Phillip Isola. What makes for good views for contrastive learning? NeurIPS, 2020.

---

### Official Review · Reviewer_DXHp · 2022-10-27

**Confidence:** 4
**Clarity, Quality, Novelty And Reproducibility:** The paper is clear, of high quality a…
**Correctness:** 3
**Technical Novelty And Significance:** 2
**Empirical Novelty And Significance:** 3
**Recommendation:** 6

**Strength And Weaknesses:**

S

+ extensive experimentation and ablation tests
+ strong results against baselines

W

+ lack of comparisons with already cited works (NCL)


**Summary Of The Paper:**

This paper proposes an unsupervised domain adaptation model for time series combining nearest-neighbourhood contrastive learning and adversarial learning.

**Summary Of The Review:**

The paper proposes a comprehensive method for unsupervised domain adaptation with strong results against baselines.

---

> ### Author Response · Authors · 2022-11-14
> **Response to Reviewer DXHp**
>
> Thank you for your time and valuable feedback that helps us improve our submission. Specifically, your comments helped us to emphasize the importance of choosing an effective CL approach in time-series UDA. Below, we discuss how we have improved our work as a result.
>
> ## Responses to “Weaknesses”
>
> **lack of comparisons with already cited works (NCL)**
>
> Thank you for your comment and for giving us the opportunity to compare our work with NCL [1]. Following your feedback, we included a new ablation study where we replaced our contrastive learning component with the one introduced in NCL (see new Appendix K.2). However, this yielded inferior results. The reason is the following: NCL considers different time windows (i.e., segments) of the same subject as positive pairs, and thus leads to a reduced prediction performance when these time windows belong to different labels (such as different activities in the sensor dataset or different decomposition labels for the healthcare dataset).
>
> We would like to further highlight that our problem setting is fundamentally different from NCL. Specifically, NCL focuses on a transfer learning setting where (phase 1) the model is first pretrained on the source dataset and (phase 2) then further fine-tuned on the **labeled** target dataset. Compared to that, in our work, we consider unsupervised domain adaptation setting where our CLUDA is jointly trained with source and target domain data in an end-to-end fashion. As such, we don’t require any labels of the target domain.
>
> ## References
>
> [1] Hugo Yèche, Gideon Dresdner, Francesco Locatello, Matthias Hüser, and Gunnar Rätsch. Neighborhood contrastive learning applied to online patient monitoring. In ICML, 2021.

---

### Public Comment · ~Vilma_Bertram1 · 2022-11-06
**Public comment on the model selection and the paper novelty**

**How the authors did hyper-parameter search without target labels**?

Unsupervised Domain Adaptation fundamentally assumes access to labeled source data and fully unlabelled target data. Therefore, how to select the best hyper-parameters with no labels available is still a long standing problem. Nevertheless, the authors of the CLUDA work have not clearly explained how they selected the hyper-parameters and only mentioned that I will quote from the paper appendix *'' We applied early stopping based on the method performance on validation set without the labels from the target domain.''* To the best of my knowledge, early stopping can only be applied with using the target labels, it is really worth clarifying how could the authors use early stopping on the target data without any labels? As model selection can be critical to the performance.

**As most of the UDA method applied to the feature space, methods that applied for visual application should also works well for time series data, why there can be a difference**?
The authors claimed in many venues that there are existing methods that proposed for images but cannot be extended for time series data. Nevertheless, their experiments already dispute this claim, as most of the baseline are already methods proposed for images and has been adapted to time series data.

** How different is your contextual adaptation from the neighbourhood clustering paper?**
The contextual adaptation is a key contribution of the paper as other components including adversarial and MoCO base contrasting already existing. However, the contextual adaptation approach seems to be very similar to ref[1]. Moreover, ref[1] has also been applied for time series data, which significantly question the novelty of the proposed work by using already existing technique.

[1] Yèche, Hugo, et al. "Neighborhood contrastive learning applied to online patient monitoring." International Conference on Machine Learning. PMLR, 2021.

---

> ### Author Response · Authors · 2022-11-14
> **Response to Public Comment (1/2)**
>
> Thank you so much for your interest in our work. Your feedback is important to improve our submission. We address your comments below.
>
> **How the authors did hyper-parameter search without target labels?**
>
> Thanks for your question. It is common for UDA to assume a labeled source domain and an unlabeled target domain while training the model and making it ready to deploy [1,2]. We follow the same principle in our work and thereby adhere to earlier work as well as to reflect real-world scenarios. At this point, we would like to note that some of the earlier works used the labeled validation target set for early stopping [3,4] by acknowledging that the results show the upperbound of the performance. We also believe this can show quite optimistic results. On the other hand, some works described that they used the validation loss [5,6] for the early stopping and model selection. However, it is not clear to us how they chose the best hyper-parameter setting when reporting the test results, because the weights of the different loss components (which is a hyperparameter) clearly affect the scale of overall validation loss, which make different configurations incomparable (e.g., setting the weights of some of the loss components to 0 would trivially yield a lower validation loss, which would contradict with the novelty of any proposed approach.).
>
> To alleviate the above issue and present realistic results, we used the following strategy during our experiments: We split each domain (for both sensor and medical datasets) into training, validation and test set, as described in our Appendix. For the hyperparameters regarding the model architecture (i.e., num layers, hidden dimension, etc.) and the training procedure (i.e., learning rate, batch size) given the fixed weights of loss components, we choose the model with the best validation loss (which involves labeled source domain and unlabeled target domain). To select the model across different loss weights (i.e., lambdas), we choose the one with the highest performance metric (e.g., accuracy, F1, or AUROC depending on the setting) on the labeled validation source set as our proxy. This choice is informed by the theory of learning from different domains [1] according to which the loss on the target domain is upperbounded by the loss on the source domain and some other additional terms. Hence, we are aiming for a better bound on the target domain by choosing a better performance on the source domain. Finally, we report the results on the labeled test domain (and labeled source domain for our case study), which have never been seen during the training or model selection process. We applied this strategy to all the models presented in our paper. Of note, the performance of our CLUDA framework and the other baselines can be further improved by introducing tailored risk scores.
>
> We thank you for raising the question of hyperparameter search and model selection. This gave us the opportunity to add more explanations into our paper about the experimental setting. In particular, we greatly expanded Appendix C to discuss the above process in depth.
>
> **As most of the UDA method applied to the feature space, methods that applied for visual application should also works well for time series data, why there can be a difference?**
>
> We rephrased our claim regarding the applicability of visual UDA methods for our task using time series. Such visual UDA methods are applicable and may serve as baselines, because of which we included them as baselines. Therefore, we revised our claim to state that’s these models are applicable but they don’t leverage unique time series properties. We updated our submission accordingly. Thank you for helping us improve the wording in our paper.
>
> **How different is your contextual adaptation from the neighbourhood clustering paper?**
>
> Thank you for your comment and for giving us the opportunity to compare our work with NCL.
>
> Difference from NCL.
>
> - NCL solves a different problem. Specifically, this work pretrains the model on the source domain. Then, it further fine-tunes the model based on the labeled target domain. Here, in our UDA setting, we considered the case where the labels of the target domain are not available and, hence, cannot be used as fine-tuning.

---

> > ### Author Response · Authors · 2022-11-14
> > **Response to Public Comment (2/2)**
> >
> > - The novelty of their representation learning is attributed to their neighborhood selection when implementing the contrastive learning objective. This work selects the neighbors among the different time windows of the same subject. However, especially on the sensor datasets, even the consecutive time windows can correspond to different types of activities (from walking to running, or from sitting to standing up etc.), therefore mapping them to similar points in the embedding space may result in an inferior performance. Further, such selection strategy of the neighbors would require careful shuffling of the data and the construction of the batches so that positive pairs can fall into the same batch. Under the low memory computational setting, this could lead to learning from different small subset of subjects at each training step, and to decreasing speed of convergence. In our CLUDA framework, we have the flexibility of randomly selecting both source and target domains and representing them in the same batch. Further, in our UDA setting, we are offering the contrastive learning as a bridge between the original source and target domains without any intermediate synthetic data generation step (see our new paragraph on UDA for videos in literature review), and this has not been done before. As such, our framework does not require any additional information regarding the subjects (such as their IDs and/or some static features), which offers a flexibility upon deployment. Of note, the mentioned paper encodes the static features of the patients before the contrastive learning objective, which could make the training potentially more trivial, since the neighboring samples (from the same patient) have exactly the same static features encoded.
> >
> > - The training paradigms are different in the sense that the mentioned work has the training in two stages. In comparison, we offer an end-to-end approach which could be readily deployed after the one stage of training has been completed.
> >
> > - Our experimental setup in healthcare is also more comprehensive. The mentioned work has stratified sampling at the patient level across train-val-test set, yet these splits contain similar demographic distributions (of patients) due to the large number of patients in the dataset. As a result, the model is pre-trained and fine-tuned on similar distributions. In comparison, we improve upon the earlier setup in mainly two different ways. First, we further split the dataset into several distinct age groups, by which the different groups show diverse health characteristics. We treat these different age groups as separate domains and extensively study the adaption performance across them. Second, we collected the healthcare dataset from two different health institutions, which are in different continents. These institutions have different patient demographics, different staffing, different medication guides, and different ICU admission and discharge procedures that constitute a greater variation across two domains. Therefore, we believe it is an important dimension of our work to study UDA across two health institutions. On top of that, we further study UDA across both different age groups and different institutions. To the best of our knowledge, this offers one of the greatest diversity across domains in UDA literature for time series.
> >
> > Following your feedback, we included a new ablation study where we replaced our contrastive learning component with the one introduced in NCL (see Appendix K.2). We find that our framework is clearly superior.
> >
> > We are thankful for your constructive comments and we are happy to receive the interest from the public.
> >
> > ## References
> >
> > [1] Shai Ben-David, John Blitzer, Koby Crammer, Alex Kulesza, Fernando Pereira, and Jennifer Wortman Vaughan. A theory of learning from different domains. JMLR, 79:151–175, 2010.
> >
> > [2] Yaroslav Ganin, Evgeniya Ustinova, Hana Ajakan, Pascal Germain, Hugo Larochelle, François Laviolette, Mario Marchand, and Victor Lempitsky. Domain-adversarial training of neural networks. JMLR, 17:2096–2030, 2016.
> >
> > [3] Garrett Wilson, Janardhan Rao Doppa, and Diane J Cook. Multi-source deep domain adaptation with weak supervision for time-series sensor data. In KDD, 2020.
> >
> > [4] Qiao Liu and Hui Xue. Adversarial spectral kernel matching for unsupervised time series domain adaptation. In IJCAI, 2021.
> >
> > [5] Sanjay Purushotham, Wilka Carvalho, Tanachat Nilanon, and Yan Liu. Variational recurrent adversarial deep domain adaptation. In ICLR, 2017.
> >
> > [6] Ruichu Cai, Jiawei Chen, Zijian Li, Wei Chen, Keli Zhang, Junjian Ye, Zhuozhang Li, Xiaoyan Yang, and Zhenjie Zhang. Time series domain adaptation via sparse associative structure alignment. In AAAI, 2021.

---

### Author Response · Authors · 2022-11-14
**Official Response to All Reviewers**

We thank all the reviewers for their time and their helpful comments! We took all the suggestions at heart, and made a substantial revision to our paper. We highlighted the changes in **red color** (see the rebuttal PDF for download). Our main improvements are the following:

1. We added additional experiments where we compared our work’s contrastive learning component to NCL (Yèche et al., 2021) (see Appendix K.2).

2. We have reworked our figure to visualize our CLUDA framework (see Figure 1 and Section 4.1).

3. We added new materials to explain how our CLUDA framework can be tailored towards variable-length time series (see Appendix L).

4. We extended our literature review to UDA for videos to highlight the novelty of our work (see revised Appendix A).

5. We added additional information on early stopping and hyperparameter selection (see new materials in Appendix C).

With the improvements above, we are confident that we remedied all weaknesses as a result, and are convinced that our paper is a valuable contribution to the literature and a good fit for ICLR 2023. We hope you agree.

---

> ### Comment · Reviewer_ZBbo · 2022-12-13
> **Final comments post author's response**
>
> The authors have addressed the comments effectively. There are a few outstanding issues which the authors should address in the revision(s).
>
> - I see that a recent work [1] on domain adaptation for time-series is not discussed, this work should be discussed in the related works/introduction.
> - The figures are still somewhat difficult to read. For instance, the paper does not make full use of the space in Fig. 1, 2, 3,4 and the tables. This should be an easy fix, and trust that the authors will pay close attention to this.
> - The authors should consider adding a link to the codebase for the benefit of the community, and details about reproducing the results at the least.
>
> [1] Jin, Xiaoyong, et al. "Domain adaptation for time series forecasting via attention sharing." International Conference on Machine Learning. PMLR, 2022.

---

> ### Comment · Reviewer_X1Eq · 2022-12-13
> **Final comments following author's response**
>
> I have read the author's response, and consider they have adressed the questions I had well. I will be maintaining my overall score (8).
> Of note:
> - I agree with the comments in the other response to the post titled "official response to all reviewers" (i.e. the parent of this post). The points outlined by reviewer ZBbo should be addressed.
> - I strongly feel this submission should be accepted. I understand and acknowledge some of the points highlighted by other reviewers, namely the link to DA methods in computer vision and the need to take more related works into account, among others. However, the contributions of this paper are important to the field of time-series, and I feel the points previously mentioned do not alone warrant rejecting this submission.

---

### Decision · Program_Chairs · 2023-01-20

**Decision:**

Accept: poster

**Justification For Why Not Higher Score:**

This paper still has some weaknesses, e.g., relevant works in other domains should be discussed and readability should be improved, we recommend Accept with poster, instead of spotlight or oral.

**Justification For Why Not Lower Score:**

Given that this paper is technically sound and experimental results are convincing, we would like to recommend Accept.

**Metareview: Summary, Strengths And Weaknesses:**

This paper proposed a framework called CLUDA to combine contrastive learning (CL) and unsupervised domain adaptation (UDA) for time-series data. In particular, the authors utilized MoCo based contrastive learning and adversarial learning in the CLUDA framework. The evaluation for CLUDA is also convincing.

Strengths:
1. The paper is well motivated. The proposed model is also clearly described.
2. After the rebuttal, the experiment part is quite good.

Weaknesses:
1. A recent work in ICML should be discussed, as suggested by Reviewer ZBbo.
2. Figures can be further improved for better readability.
3. Writing/wording can be further improved (try to avoid inaccurate or unsupported claims/descriptions). For example, original descriptions about visual UDA methods brought confusions for some reviewers. In addition, some methods in this sentence “Another research stream has developed time series methods for transfer learning from the source domain to the target domain” are for unsupervised representation learning, not merely for transfer learning.


**Note From Pc:**

if the above contains the word "oral" or "spotlight" please see: "oral" presentation means -> notable-top-5% and "spotlight" means -> notable-top-25%. As stated in our emails, we are disassociating presentation type from AC recommendations

**Summary Of Ac-Reviewer Meeting:**

The AC-reviewer meeting was held on 13 Dec. Both reviewers ZBbo and X1Eq joined the meeting, while xEBG giving a rating of 3 is on holiday and thus not able to join the meeting.

In fact three of us are quite supporting to accept this paper. ZBbo mentioned that this paper should be accepted after some minor issues are addressed. X1Eq agreed with review comments that some contrastive UDA methods in computer vision should be discussed. He also argued that some points from xEBG are not sufficient to reject a paper. After the discussion, both ZBbo and X1Eq updated the posts on OpenReview.